# Flatter, faster: scaling momentum for optimal speedup of SGD

## Abstract

Commonly used optimization algorithms often show a trade-off between good generalization and fast training times. For instance, stochastic gradient descent (SGD) tends to have good generalization; however, adaptive gradient methods have superior training times. Momentum can help accelerate training with SGD, but so far there has been no principled way to select the momentum hyperparameter. Here we study training dynamics arising from the interplay between SGD with label noise and momentum in the training of overparametrized neural networks. We find that scaling the momentum hyperparameter $1 - \beta$ with the learning rate to the power of $2/3$ maximally accelerates training, without sacrificing generalization. To analytically derive this result we develop an architecture-independent framework, where the main assumption is the existence of a degenerate manifold of global minimizers, as is natural in overparametrized models. Training dynamics display the emergence of two characteristic timescales that are well-separated for generic values of the hyperparameters. The maximum acceleration of training is reached when these two timescales meet, which in turn determines the scaling limit we propose. We confirm our scaling rule for synthetic regression problems (matrix sensing and teacher-student paradigm) and classification for realistic datasets (ResNet-18 on CIFAR10, 6-layer MLP on FashionMNIST), suggesting the robustness of our scaling rule to variations in architectures and datasets.

## 1 Introduction

The modern paradigm for optimization of deep neural networks has engineers working with vastly overparametrized models and training to near perfect accuracy (Zhang et al., 2017). In this setting, a model will typically have not just isolated minima in parameter space, but a continuous set of minimizers, not all of which generalize well. Liu et al. (2020) demonstrate that depending on parameter initialization and hyperparameters, stochastic gradient descent (SGD) is capable of finding minima with wildly different test accuracies. Thus, the power of a particular optimization method lies in its ability to select a minimum that generalizes amongst this vast set. In other words, good generalization relies on the *implicit bias or regularization* of an optimization algorithm.

There is a significant body of evidence that training deep nets with SGD leads to good generalization. Intuitively, SGD appears to prefer flatter minima (Keskar et al., 2017; Wu et al., 2018; Xie et al., 2020), and flatter minima generalize better (Hochreiter & Schmidhuber, 1997). More recently, a variant of SGD which introduces "algorithmic" label noise has been especially amenable to rigorous treatment. In the overparametrized setting Blanc et al. (2020) were able to rigorously determine that SGD with label noise converges not just to any minimum, but to those minima that lead to the smallest trace norm of the Hessian. However, Li et al. (2022) show that the dynamics of this regularization happen on a timescale proportional to the inverse square of the learning rate $\eta$ – much slower than the time to first converge to an interpolating solution. Therefore we consider the setting where we remain near the local minima, which is responsible for significant regularization after the initial convergence of the train loss (Blanc et al., 2020).

With the recent explosion in size of both models and datasets, training time has become an important consideration in addition to asymptotic generalization error. In this context, adaptive gradient methods such as Adam (Kingma & Ba, 2015) are unilaterally preferred over variants of SGD, even though they often yield worse generalization errors in practical settings (Keskar & Socher, 2017;

Wilson et al., 2017), though extensive hyperparameter tuning (Choi et al., 2019) or scheduling (Xie et al., 2022) can potentially obviate this problem.

These two constraints motivate a careful analysis of how momentum accelerates SGD. Classic work on acceleration methods, which we refer to generally as momentum, have found a provable benefit in the deterministic setting, where gradient updates have no error. However, rigorous guarantees have been harder to find in the stochastic setting, and remain limited by strict conditions on the noise (Polyak, 1987; Kidambi et al., 2018) or model class and dataset structure Lee et al. (2022).

In this work, we show that there exists a scaling limit for SGD with momentum (SGDM) which provably increases the rate of convergence.

**Notation.** In what follows, we denote by $C^n$, for $n = 0, 1, \ldots$ the set of functions with continuous $n^{\text{th}}$ derivatives. For any function $f$, $\partial f[u]$ and $\partial^2 f[u, v]$ will denote directional first and second derivatives along directions defined by vectors $u, v \in \mathbb{R}^D$. We may occasionally also write $\partial^2 f[\Sigma] = \sum_{i,j=1}^{D} \partial^2 f[e_i, e_j] \Sigma_{ij}$. Given a submanifold $\Gamma \subset \mathbb{R}^D$ and $w \in \Gamma$, we denote by $T_w \Gamma$ the tangent space to $\Gamma$ in $w$, and by $P_L(w)$ the projector onto $T_w \Gamma$ (we will often omit the dependence on $w$ and simply write $P_L$). Given a matrix $H \in \mathbb{R}^D \times \mathbb{R}^D$, we will denote by $H^\top$ the transpose, and $H^\dagger$ the pseudoinverse of $H$.

## 1.1 HEURISTIC EXPLANATION FOR OPTIMAL MOMENTUM-BASED SPEEDUP

Deep neural networks typically posses a manifold of parametrizations with zero training error. Because the gradients of the loss function vanish along this manifold, the dynamics of the weights is completely frozen under gradient descent. However, as appreciated by Wei & Schwab (2019) and Blanc et al. (2020), noise can generate an average drift of the weights along the manifold. In particular, SGD noise can drive the weights to a lower-curvature region which, heuristically, explains the good generalization properties of SGD. Separately, it is well-known that adding momentum typically leads to acceleration in training Sutskever et al. (2013) Below, we will see that there is a nontrivial interplay between the drift induced by noise and momentum, and find that acceleration along the valley is maximized by a particular hyperparameter choice in the limit of small learning rate. In this section we will illustrate the main intuition leading to this prediction using heuristic arguments, and defer a more complete discussion to Sec. 3.

We model momentum SGD with label noise using the following formulation

$$\pi_{k+1} = \beta \pi_k - \nabla L(w_k) + \epsilon \sigma(w_k) \xi_k, \qquad w_{k+1} = w_k + \eta \pi_{k+1}, \tag{1}$$

where $\eta$ is the learning rate, $\beta$ is the (heavy-ball) momentum hyperparameter (as introduced by Polyak (1964)), $w \in \mathbb{R}^D$ denotes the weights and $\pi \in \mathbb{R}^D$ denotes momentum (also called the auxiliary variable). $L : \mathbb{R}^D \to \mathbb{R}$ is the training loss, and $\sigma : \mathbb{R}^D \to \mathbb{R}^{D \times r}$ is the noise function, whose dependence on the weights allows to model the gradient noise due to SGD and label noise. Specifically this admits modeling phenomena such as automatic variance reduction (Liu & Belkin, 2020), and expected smoothness, satisfying only very general assumptions such as those developed by Khaled & Richtárik (2020). Finally, $\xi_k \in \mathbb{R}^r$ is sampled i.i.d. at every timestep $k$ from a distribution with zero mean and unit variance, and $\epsilon > 0$.

We will now present a heuristic description of the drift dynamics that is induced by the noise along a manifold of minimizers $\Gamma = \{w : L(w) = 0\} \subseteq \mathbb{R}^D$, in the limit $\epsilon \to 0$. In practice, this limit corresponds to choosing small strength of label noise and large minibatch size. Let us assume that the weights at initial time $w_0$ are already close to $\Gamma$, and $\pi_0 = 0$. Because of this, the gradients of $L$ at $w_0$ are very small, and so only fluctuations transverse to the manifold will generate systematic drifts. Denoting by $\delta w_k = w_0 - w_k$ the displacement of the weights after $k$ timesteps, let us Taylor expand the first equation in (1) to get $\pi_{k+1} = \beta \pi_k - \nabla^2 L(w_0)[\delta w_k] + \epsilon \sigma(w_0) \xi_k$. By construction, the Hessian $\nabla^2 L(w_0)$ vanishes along the directions tangent to $\Gamma$, while in the transverse direction we have an Ornstein-Uhlenbeck (OU) process. The number of time steps it takes to this process to relax to the stationary state, or mixing time, is $\tau_1 = \Theta(1/(1-\beta))$ as $\beta \to 1$, which can be anticipated from the first equation in (1) since $\pi_{k+1} - \pi_k \sim -(1-\beta)\pi_k$ (see Sec. 3.3 for a more detailed derivation). After this time, the variance of this linearized OU process becomes independent of the time step, and can be estimated to be (see Appendix F): $\langle (\delta w_k^T)^\top \delta w_k^T \rangle = \Theta(\epsilon^2 \eta/(1-\beta))$, where $\langle \cdots \rangle$ denotes the noise average. To keep track of the displacements in the longitudinal directions, $\delta w_k^L$, we need to look at the cubic order in

the Taylor expansion of $L(w_0 + \delta w_k)$, i.e. $\partial^2(\nabla L)[\delta w_k, \delta w_k]$. Let $P_L$ be the projector onto the tangent space. The expectation value of momentum, upon applying the longitudinal projector, is $P_L\langle \pi_{k+1} \rangle = -P_L \sum_{j=0}^{k} \beta^{k-j} \langle \nabla L(w_j) \rangle = -\frac{1}{2} \sum_{j=0}^{k} \beta^{k-j} \partial^2 (P_L \nabla L)[\langle (\delta w_j)^\top \delta w_j \rangle] = -\frac{1}{2} \frac{1}{1-\beta} \partial^2 (P_L \nabla L)[\langle (\delta w_j)^\top \delta w_j \rangle]$. The variance of the transverse displacements therefore generates longitudinal motion, and the latter will then scale as $P_L \langle \pi_{k+1} \rangle = \Theta\left(\epsilon^2 \eta / (1-\beta)^2\right)$. Using the second equation in (1), we see that $P_L(\langle \delta w_{k+1} \rangle - \langle \delta w_k \rangle) = P_L \langle \eta \pi_{k+1} \rangle$. Define $\tau_2$ to be the number of time steps it takes so that the displacement of $w_k$ along $\Gamma$ becomes a finite number as we take $\epsilon \to 0$ first, and $\eta \to 0$ afterward. From the above considerations, we then find that $\tau_2 = \Theta\left((1-\beta)^2 / \epsilon^2 \eta^2\right)$.

The key observation now is that, when the weights are initialized near the valley, the convergence time is controlled by the largest timescale among $\tau_1$ and $\tau_2$. While $\tau_2$ decreases as $\beta \to 1$, $\tau_1$ increases. Therefore, the optimal speedup of training is achieved when these two timescales intersect $\tau_1 = \tau_2$, which happens for $1 - \beta = C\eta^{2/3}$. More generally, we consider a double scaling where $\beta \to 1$ as $\eta \to 0$ according to

$$\beta = 1 - C\eta^\gamma. \tag{2}$$

We consider this scaling limit in this paper, and we thus find that the scaling power $\gamma = 2/3$ achieves the optimal speed-up along the zero-loss manifold.

## 1.2 Limit Drift-Diffusion

We now describe the rationale for obtaining the limiting drift-diffusion on the zero-loss manifold, for a process of the form (1) which foreshadows the rigorous results presented in section 3. As discussed above, the motion along the manifold is slow, as it takes $\Theta(\epsilon^{-2})$ time steps to have a finite amount of longitudinal drift. We want to extract this slow longitudinal sector of the dynamics by projecting out the fast-moving components of the weights. For stable values of the optimization hyperparameters, the noiseless ($\epsilon^2 = 0$) dynamics (1) will map a generic pair $(\pi, w)$, as $k \to \infty$, to $(0, w_\infty)$, where $w_\infty \in \Gamma$. Define $\Phi : \mathbb{R}^{D \times D} \to \mathbb{R}^D$ to be this mapping, i.e. $\Phi(\pi, w) = w_\infty$. As we now show, when $\epsilon > 0$, $\Phi$ can be used precisely to project onto the slow, noise-induced longitudinal dynamics. Let us collectively denote $x_k = (\pi_k, w_k)$ and write eq. (1) as $x_{k+1} = x_k + F(x_k) + \epsilon \tilde{\sigma}(x_t) \xi_t$. We can perform a Taylor expansion in $\epsilon$ to obtain $\Phi(x_{t+1}) - \Phi(x_t) = \partial \Phi(x_t)[\epsilon \tilde{\sigma}(x_t) \xi_t] + \partial^2 \Phi(x_t)[\epsilon \tilde{\sigma}(x_t) \xi_t, \epsilon \tilde{\sigma}(x_t) \xi_t] + \cdots$. Therefore, denoting $Y(t = \epsilon^2 k) = \Phi(x_k)$, the limit dynamics as $\epsilon \to 0$ can be well-approximated by the continuous time equation

$$dY = \partial \Phi(x_t)[\tilde{\sigma}(x_t) \xi_t]\sqrt{dt} + \partial^2 \Phi(x_t)[\tilde{\sigma}(x_t) \xi_t, \tilde{\sigma}(x_t) \xi_t]dt, \tag{3}$$

where we interpret the time increment $dt = \epsilon^2$, and $\langle \xi_t^2 \rangle = 1$. Note that until here we have not taken a small learning rate limit. The learning rate can be finite, as far as the map $\Phi(\pi, x)$ exists. The small noise limit is sufficient to allow a continuous-time description of the limit dynamics because the noise-induced drift-diffusion along the valley requires $\Theta(\epsilon^{-2})$ timesteps to lead to appreciable longitudinal displacements.

A similar approach to what we just described was used in Li et al. (2022), although in our case the limit drift-diffusion is obtained in the small noise limit, rather than small learning rate. The reason for our choice is that, since we scale $\beta$ according to (2), the deterministic part of eq. (1) becomes degenerate as we take $\eta \to 0$, in which case it would not possible to apply the mathematical framework of Katzenberger (1991) on which our results below rely. To further simplify our analysis, particularly the statement of Theorem B.4, we will further take $\eta \to 0$ after taking $\epsilon \to 0$, and retain only leading order contributions in $\eta$.

The **main contributions** of this paper are:

1. We develop a general formalism to study SGD with (heavy-ball) momentum in Sec. 3, extending the framework of Li et al. (2022) to study convergence rates and generalization with momentum.

2. We find a novel scaling regime of the momentum hyperparameter $1 - \beta \sim \eta^\gamma$, and demonstrate a qualitative change in the noise-induced training dynamics as $\gamma$ is varied.

3. We identify a special scaling limit, $1 - \beta \sim \eta^{2/3}$, where training achieves a *maximal speedup at fixed learning rate $\eta$*.

4. In Sec. 4, we demonstrate the relevance of our theory with experiments on toy models (2-layer neural networks, and matrix sensing) as well as realistic models and datasets (ResNet-18 on CIFAR10).

## 2 RELATED WORKS

**Loss Landscape in Overparametrized Networks** The geometry of the loss landscape is very hard to understand for real-world models. Choromanska et al. (2015) conjectured, based on empirical observations and on an idealized model, that most local minima have similar loss function values. Subsequent literature has shown in wider generality the existence of a manifold connecting degenerate minima of the loss function, particularly in overparametrized models. This was supported by work on mode connectivity (Freeman & Bruna, 2017; Garipov et al., 2018; Draxler et al., 2018; Kuditipudi et al., 2019), as well as on empirical observations that the loss Hessian possesses a large set of (nearly) vanishing eigenvalues (Sagun et al., 2016; 2017). In particular, Nguyen (2019) showed that for overparametrized networks with piecewise linear activations, all global minima are connected within a unique valley.

**The Implicit Regularization of SGD** Wei & Schwab (2019), assuming the existence of a zero-loss valley, observed that SGD noise leads to a decrease in the trace of the Hessian. Blanc et al. (2020) demonstrated that SGD with label noise in the overparametrized regime induces a regularized loss that accounts for the decrease in the trace of the Hessian. Damian et al. (2021) extend this analysis to finite learning rate. HaoChen et al. (2021) study the effect of non-isotropic label noise in SGD and find a theoretical advantage in a quadratic overparametrized model. Wu et al. (2022) show that only minima with small enough Hessians (in Frobenius norm) are stable under SGD. The specific regularization induced by SGD was found in quadratic models (Pillaud-Vivien et al., 2022), 2-layer Relu networks (Blanc et al., 2020), linear models (Li et al., 2022), diagonal networks (Pesme et al., 2021). Additionally, Kunin et al. (2021) and Xie et al. (2021) studied the diffusive dynamics induced by SGD both empirically and in a simple theoretical model.

**Momentum in SGD and Adaptive Algorithms** Momentum is a general term applied to techniques introduced to speed up gradient descent. Popular implementations include Nesterov (Nesterov, 1983) and Heavy Ball (HB) or Polyak (Polyak, 1964). We focus on the latter in this paper, which we refer to simply as momentum. Momentum provably improves convergence time in the deterministic setting. Intuitively, introducing $\beta$ in 1 gives the motion in parameter space an effective "inertia" or memory, which promotes motion not strictly following the local gradient, but moving rather along the directions which persistently decrease the loss function across iterations Polyak (1964); Sutskever et al. (2013). Less is known rigorously when stochastic gradient updates are used. Indeed, Polyak (1987) suggests the benefits of acceleration with momentum disappear with stochastic optimization unless certain conditions are placed on the properties of the noise. See also (Jain et al., 2018; Kidambi et al., 2018) for more discussion and background on this issue. Nevertheless, in practice it is widely appreciated that momentum is important for convergence and generalization (Sutskever et al., 2013), and widely used in modern adaptive gradient algorithms Kingma & Ba (2015). Some limited results have been obtained showing speedup in the mean-field approximation (Mannelli & Urbani, 2021) and linear regression Jain et al. (2018). Modifications to Nesterov momentum to make it more amenable to stochasticity (Liu & Belkin, 2020; Allen-Zhu, 2017), and near saddle points (Xie et al., 2022) have also been considered.

## 3 THEORETICAL RESULTS

### 3.1 GENERAL SETUP

Following the line from section 1.2 In this and the following section, we will rigorously derive the limiting drift-diffusion equation for the weights on the zero-loss manifold $\Gamma$, and extract the timescale $\tau_2$ associated to this noise-induced motion. In Sec. 3.3 we will then compare $\tau_2$ to the timescale $\tau_1$ associated to the noiseless dynamics and evaluate the optimal value of $\gamma$ discussed around Eq. (2). We will use Eq. (1) to model momentum SGD. As illustrated in Sec. 1.1, the drift is controlled by the second moment of fluctuations, and we thus expect the drift timescale to be $\Theta(\epsilon^2)$.

We will then rescale time $k = t/\epsilon^2$, so that the motion in the units of $t$ is $O(1)$ as $\epsilon \to 0$. More explicitly, take $\epsilon_n$ to be a positive sequence such that $\epsilon_n \to 0$ as $n \to \infty$. For each $n$ we consider the stochastic process that solves Eq. (1):

$$X_n(t) = X_n(0) + \int_0^t \tilde{\sigma}(X_n)dZ_n + \int_0^t F(X_n)dA_n \,, \tag{4}$$

with

$$A_n(t) = \left\lfloor \frac{t}{\epsilon_n^2} \right\rfloor, \qquad Z_n(t) = \epsilon_n \sum_{k=1}^{A_n(t)} \xi_k \tag{5}$$

and where $X(t = \epsilon^2 k) = (\pi_k, w_k)$, $\tilde{\sigma}(X) = (\sigma, \eta\sigma)$ and $F(X) = ((\beta - 1)\pi - \nabla L(w), \eta(\beta\pi - \nabla L(w)))$. $\lfloor x \rfloor$ denotes the integer part of a real number $x$. See Appendix B.2 for a proof of equivalence between (1) and (4).

**Assumption 3.1.** *The loss function $L : \mathbb{R}^D \to \mathbb{R}$ is a $C^3$ function whose first 3 derivatives are locally Lipschitz, $\sigma$ is continuous, and $\Gamma = \{w \in \mathbb{R}^D : L(w) = 0\}$ is a $C^2$-submanifold of $\mathbb{R}^D$ of dimension $M$, with $0 \leq M \leq D$. Additionally, for $w \in \Gamma$, $rank(\nabla^2 L(w)) = D - M$.*

**Assumption 3.2.** *There exists an open neighborhood $U$ of $\{0\} \times \Gamma \subseteq \mathbb{R}^D \times \mathbb{R}^D$ such that the gradient descent starting in $U$ converges to a point $x = (\pi, w) \in \{0\} \times \Gamma$. More explicitly, for $x \in U$, let $\psi(x, 0) = x$ and $\psi(x, k+1) = \psi(x, k) + F(\psi(x, k))$, i.e. $\psi(x, k)$ is the $k^{th}$ iteration of $x + F(x)$. Then $\Phi(x) \equiv \lim_{k\to\infty} \psi(x, k)$ exists and is in $\Gamma$. As a consequence, $\Phi \in C^2$ on $U$ (Falconer, 1983).*

### 3.2 Limiting drift-diffusion in momentum SGD

In this section we shall obtain the explicit expression for the limiting drift-diffusion. The general framework is based on Katzenberger (1991) (reviewed in Appendix B). Before stating the result, we will need to introduce a few objects.

**Definition 3.3.** For a symmetric matrix $H \in \mathbb{R}^D \times \mathbb{R}^D$, and $W_H = \{\Sigma \in \mathbb{R}^D \times \mathbb{R}^D : \Sigma = \Sigma^\top, HH^\dagger\Sigma = H^\dagger H\Sigma = \sigma\}$, we define the operator $\tilde{\mathcal{L}}_H : W_H \to W_H$ with $\tilde{\mathcal{L}}_H S \equiv \{H, S\} + \frac{1}{2}C^{-2}\eta^{1-2\gamma}[[S, H], H]$, with $[S, H] = SH - HS$. It can be shown that the operator $\tilde{\mathcal{L}}_H$ is invertible (see Lemma C.3).

Consider the process in Eq. (4). Note that, while at initialization we can have $X_n(0) \notin \Gamma$, the solution $X_n(t) \to \Gamma$ as $n \to \infty$, i.e. it becomes discontinuous. This is an effect of the speed-up of time introduced around Eqs. (4),(5). To overcome this issue, it is convenient to introduce $Y_n(t) \equiv X_n(t) - \psi(X_n(0), A_n(t)) + \Phi(X_n(0))$, so that $Y_n(0) \in \Gamma$ is initialized on the manifold.

**Theorem 3.4** (Informal). *Suppose the loss function $L$, the noise function $\sigma$, the manifold of minimizers $\Gamma$ and the neighborhood $U$ satisfy assumptions (3.1) and (3.2), and that $X_n(0) = X(0) \in U$. Then, as $\epsilon_n \to 0$, and subsequently taking $\eta \to 0$, $Y_n(t)$ converges to $Y(t)$, where the latter satisfies the limiting drift-diffusion equation*

$$\begin{aligned} dY =&(\tfrac{1}{C}\eta^{1-\gamma} + \eta)P_L\sigma dW - \tfrac{1}{2C^2}\eta^{2-2\gamma}(\nabla^2 L)^\dagger\partial^2(\nabla L)[\Sigma_{LL}]dt \\ &- \tfrac{1}{C^2}\eta^{2-2\gamma}P_L\partial^2(\nabla L)[(\nabla^2 L)^\dagger\Sigma_{TL}]dt - \tfrac{1}{2C^2}\eta^{2-2\gamma}P_L\partial^2(\nabla L)[\tilde{\mathcal{L}}_{\nabla^2 L}^{-1}\Sigma_{TT}]dt \,, \end{aligned} \tag{6}$$

*where $W(t)$ is a Wiener process.*

A rigorous version of this theorem is given in section B. The first term in Eq. (6) induces diffusion in the longitudinal direction. The second term is of geometrical nature, and is necessary to guarantee that $Y(t)$ remains on $\Gamma$. The second line describes the drift induced by the transverse fluctuations.

Eq. (6) resembles in form that found in Li et al. (2022), although there are two crucial differences. First, time has been rescaled using the strength of the noise $\epsilon$, rather than the learning rate. The different rescaling was necessary as the forcing term $F$ in Eq. (4) depends non-homogeneously on $\eta$, and thus the theory of Katzenberger (1991) would not be directly applied had we taken the small learning rate limit. Second, and more crucially, the drift terms in Eq. (6) are proportional to $\eta^{2-2\gamma}$, which is a key ingredient leading to the change in hierarchy of the timescales discussed in Sec. 1.1. One final difference, is that the last term involves the operator $\tilde{\mathcal{L}}_H$ instead of the Lyapunov operator.

For $\gamma < \frac{1}{2}$, $\tilde{\mathcal{L}}_H$ reduces to the Lyapunov operator $\mathcal{L}_H$ at leading order in $\eta$, with $\mathcal{L}_H S \equiv \{H, S\}$. For $\gamma > \frac{1}{2}$, however, we cannot neglect the $\eta$-dependent term in $\tilde{\mathcal{L}}_H$ (see discussion at the end of Appendix C).

**Corollary 3.5.** *In the case of label noise, i.e. when, for $w \in \Gamma$, $\Sigma = c\nabla^2 L$ , for some constant $c > 0$, Eq. (6) reduces to*

$$dY = -\frac{\epsilon^2 \eta^{2-2\gamma}}{4C^2} P_L \nabla \, Tr(c\nabla^2 L) dt \,, \tag{7}$$

*where we have rescaled time back to $t = k$, i.e. we performed $t \to t\epsilon^2$.*

### 3.3 SEPARATION OF TIMESCALES AND OPTIMAL MOMENTUM SCALING

The above results provide the estimate for the timescale $\tau_2$ of the drift along the zero-loss valley. As discussed in Sec. 1.1, training along the zero-loss manifold $\Gamma$ is maximally accelerated if this time scale is equal to the timescales $\tau_1$ for relaxation of off-valley perturbations. As we take $\epsilon \to 0$, this relaxation is governed by the nonzero eigenvalues of the Hessian as well as by the learning rate $\eta$ and momentum $\beta$. Therefore we expect $\tau_1 = \Theta(\epsilon^0)$, and this will be confirmed by the analysis below. It will be therefore sufficient to obtain the leading order expression of $\tau_1$ by focusing on the noiseless $\epsilon = 0$ dynamics. Additionally, since we are interested in local relaxation, it will suffice to look at the linearized dynamics around $\Gamma$.

Working in the extended phase space $x_k = (\pi_k, w_k)$, and linearizing Eq. (1) around a fixed point $x^* = (0, w^*)$, with $w^* \in \Gamma$, the linearized update rule is $\delta x_{k+1} = J(x^*) \delta x_k$, where $\delta x_k = x_k - x^*$ and $J(x^*)$ is the Jacobian evaluated at the fixed point (with the explicit form given in Eq. (133)). Denote by $q^i$ the eigenvector and $\lambda_i$ the corresponding eigenvalue of the Hessian. We show in Appendix E that the Jacobian is diagonalized by the eigenvectors $k_\pm^i = (\mu_\pm^i q^i, q^i)$ with eigenvalues

$$\kappa_\pm^i = \frac{1}{2} \left( 1 + \beta - \eta\lambda_i \pm \sqrt{(1 + \beta - \eta\lambda_i)^2 - 4\beta} \right), \tag{8}$$

and $\mu_\pm^i = \beta\eta\kappa_\pm^i - 1 - \eta\lambda_i$. We proceed to study the decay rate of the different modes of the Jacobian to draw conclusions about the characteristic timescales of fluctuations around the valley.

**Longitudinal motion:** On the valley, the Hessian will have a number of "zero modes" with $\lambda_i = 1$. These lead to two distinct modes in the present setting with momentum, which we distinguish as pure and mixed. The first pure longitudinal mode is an exact zero mode which has $\kappa_+^i = 1$ with $k_+^i = (0, q^i)$, corresponding to translations of the parameters along the manifold, and keeping $\pi = 0$ at its fixed point value. The second mode is a mixed longitudinal mode with $\kappa_-^i = \beta$ with $k_-^i = (-(1 - \beta)/(2\beta\eta)q^i, q^i)$. This mode has a component of $\pi$ along the valley, which must subsequently decay because the equilibrium is a single point $\pi = 0$. Therefore, this mode decays at the characteristic rate $\beta$ for $\pi$, gleaned directly from Eq. (1).

**Transverse motion:** When the $w$ and $\pi$ are perturbed along the transverse directions $q^i$ with positive $\lambda_i$, the relaxation behavior exhibits a qualitative change depending on $\beta$. Using the scaling function $\beta = 1 - C\eta^\gamma$, for small learning rate, the spectrum is purely real for $\gamma < 1/2$, and comes in complex conjugate pairs for $\gamma > 1/2$. This leads to two distinct scaling behaviors for the set of timescales. Defining a positive $c_1 \le \min\{\lambda_i | \lambda_i > 0\}$, we find: 1) For $\gamma < 1/2$, transverse modes are purely decaying as $(1 - C\eta^\gamma)^k \le |\delta x_k^T| \le (1 - (c_1/C)\eta^{1-\gamma})^k$, with the lower bound set by the mixed longitudinal mode. For $\gamma > 1/2$, the transverse modes are oscillatory but with an envelope that decays like $|\delta x_k^{T,env}| \approx (1 - C\eta^\gamma)^{k/2}$. We leave the derivation of these results to Appendix (E).

Collecting these results, we can describe the hierarchy of timescales $\tau_1$ in the deterministic regime as a function of $\gamma$ (excluding the pure longitudinal zero mode):

| $\tau_1^{-1}$ | $\gamma < 1/2$ | $\gamma > 1/2$ |
|---|---|---|
| Long. | $\eta^\gamma$ | $\eta^\gamma$ |
| Transv. | $\eta^{1-\gamma}, \eta^\gamma$ | $\eta^\gamma, \eta^\gamma$ |

These are illustrated schematically in Fig. 1(a), where the finite timescales are shown as a function of $\gamma$. We compare these "equilibration" timescales $\tau_1$, i.e. characteristic timescales associated with

relaxation back to the zero-loss manifold, with the timescale $\tau_2 \sim \eta^{2(\gamma-1)}$ associated with drift-diffusion of the noise-driven motion along the zero-loss manifold Eq. (6). For small $\gamma$, the timescale associated with the drift-diffusion along the valley is much faster than that associated with the relaxation of the dynamics toward steady state. Transverse and mixed longitudinal fluctuations relax much faster than the motion along the valley, and produce an effective drift toward the minimizer of the implicit regularizer. However, the timescales collide at $\gamma = 2/3$, suggesting a transition to a qualitatively different transport above this value, where the transverse and the mixed longitudinal dynamics, having a long timescale, will disrupt the longitudinal drift Eq. (6). This leads us to propose $\gamma = \frac{2}{3}$ as the optimal choice for SGD training. We consistently find evidence for such a qualitative transition in our experiments below. In addition, we see that speedup of SGD with label noise is in fact maximal at this value where the timescales meet.

## 3.4 A SOLVABLE EXAMPLE

In this section we analyse a model that will allow us to determine, on top of the optimal exponent $\gamma = \frac{2}{3}$, also the prefactor $C$. We will specify to a 2-layer linear MLP, which is sufficient to describe the transition in the hierarchy of timescales described above, and is simple enough to exactly compute $C$. We will show in Sec. 4.1 that $C$ depends only mildly on the activation function. We apply a simple matching principle to determine $C$, by asking that the deterministic timescale $\tau_1$ is equal to the drift-diffusion timescale $\tau_2$. In the previous section, we found the critical $\gamma = 2/3$ by requiring these timescales have the same scaling in $\eta$. In order to determine $C$, we need more details of the model architecture.

**Definition 3.6** (UV model). We define the UV model as a 2-layer linear network parametrized by $f(x) = \frac{1}{\sqrt{n}}UVx \in \mathbb{R}^m$, where $x \in \mathbb{R}^d$, $V \in \mathbb{R}^{n \times d}$, and $U \in \mathbb{R}^{m \times n}$. For $d = m = 1$, we refer to this as the **vector UV model** (Rennie & Srebro, 2005; Saxe et al., 2014; Lewkowycz et al., 2020).

For a training dataset $\mathcal{D} = \{(x^a, y^a) | a = 1, ..., P\}$, the dataset covariance matrix is $\Sigma_{ij} = \frac{1}{P}\sum_{a=1}^{P} x_i^a x_j^a$, and the dataset variance is $\mu_2 = \text{tr}\Sigma$. For mean-squared error loss, it is possible to explicitly determine the trace of the Hessian (see Appendix D). SGD with label noise introduces $y^a \to y^a + \epsilon\xi_t$ where $\langle\xi_t^2\rangle = 1$, from which we identify $\sigma_{\mu,ja} = P^{-1}\nabla_\mu f_j(x^a)$, where $\mu$ runs over all parameter indices, $j \in [m]$, and $a \in [P]$. With this choice, the SGD noise covariance satisfies $\sigma\sigma^T = P^{-1}\nabla^2 L$. Equipped with this, we may use Corollary 3.5 with $c = P^{-1}$ to determine the effective drift (presented in Appendix D). For the vector UV model, the expression simplifies to $dY = -\tau_2^{-1}Y\,dt$, with

$$\tau_2^{-1} = \frac{\eta^{2-2\gamma}\epsilon^2\mu_2}{2nPC^2} . \tag{9}$$

The timescale of the fast initial phase $\tau_1^{-1} = (C/2)\eta^\gamma$ follows from the previous section. Then requiring $\tau_1 = \tau_2$ implies not only $\gamma = 2/3$, but

$$C = \left(\frac{\epsilon^2\mu_2}{Pn}\right)^{1/3} . \tag{10}$$

One particular feature to note here is that $C$ will be small for overparametrized models and/or training with large datasets.

## 4 EXPERIMENTAL VALIDATION

### 4.1 2-LAYER MLP WITH LINEAR AND NON-LINEAR ACTIVATIONS

The first experiment we consider is the vector UV model analyzed in Sec. 3.4. Our goal with this experiment is to analyze a simple model, and show quantitative agreement with theoretical expectations. Though simple, this model shows almost all of the important features in our analysis: the $2(1-\gamma)$ exponent below $\gamma = \frac{2}{3}$, the $\gamma$ exponent above $\frac{2}{3}$, and the constant $C$ theoretically evaluated in Sec. 3.4.

To this end, we extract the timescales at different values of $\gamma$ and show them in Fig. 1. We train on an artificially generated dataset $\mathcal{D} = \{(x^a, y^a)\}_{a=1}^{5}$ with $x \sim \mathcal{N}(0, 1)$ and $y = 0$. We use the

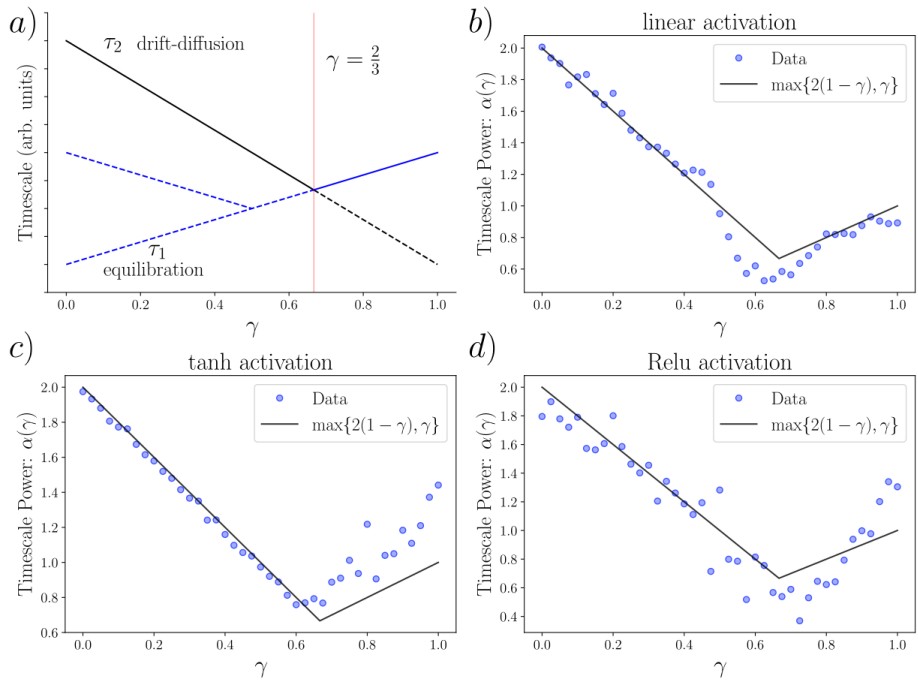

Figure 1: Timescale of training as a function of $\gamma$. $a$): theoretical prediction with blue line representing the timescale for equilibration while the black line shows the timescale of the drift along $\Gamma$. The maximum of these two gives the overall timescale. In $b$), $c$), and $d$) we demonstrate this result with the vector $UV$ model with linear, $\tanh$ and $\mathrm{Relu}$ activations respectively.

full dataset to train. From Eq.(124) we know that the norm of the weights follows an approximately exponential trajectory as it approaches the widest minimum ($U = V = 0$). We therefore measure convergence timescale, $T_c$, by fitting an exponential $ae^{-t/T_c}$ to the squared distance from the origin, $|\mathbf{U}|^2 + |\mathbf{V}|^2$. To extract the scaling of $T_c$ with $\gamma$ we perform SGD label noise with learning rates $\eta \in [10^{-3}, 10^{-1}]$ and corresponding momentum parameters $\beta = 1 - C\eta^\gamma$. We fit the timescale to a power-law in the learning rate $T_c(\eta, \gamma) = T_0\eta^{-\alpha(\gamma)}$ (see Fig. 1(b)). Imposing that $T_0$ be independent of $\gamma$, as predicted by theory, we found the numerical value $C \approx 0.2$, which is consistent with the theoretical estimate of $C = 0.17$ from Sec. 3.4. We find consistency with prediction across all the values of $\gamma$ we simulated. Note that, for $\gamma > \frac{2}{3}$ the timescale estimate fluctuates more which is a consequence of having a slower timescale for the transverse modes. As discussed at the end of Sec. 3.3, such slowness disrupts the drift motion along the manifold. $\gamma = \frac{2}{3}$ is clearly the optimal scaling.

We repeated the same experiments using nonlinear activations, specifically we considered tanh and ReLU acting on the first layer. The timescales for tanh are shown in Fig. 1(c), and we refer the reader to Appendix A for the ReLU case. The optimal scaling value is still $\gamma = \frac{2}{3}$ for both tanh and ReLU and the optimal $C$ remains close to 0.2.

## 4.2 RESNET18 ON CIFAR10

We now verify our predictions on a realistic problem, which will demonstrate the robustness of our analysis. We focus on ResNet18 (He et al., 2016), specifically implemented by Liu (2021), classifier trained on CIFAR10 Krizhevsky et al. (2009). We aim to extrapolate the theory by showing optimal acceleration with our hyperparameter choice once training reaches an interpolating solution. To this purpose, we initialize the network on the valley, obtained starting from a random weight values and training the network using full batch gradient descent without label noise and with a fixed value $\beta = 0.9$ until it reaches perfect training accuracy. With this initialization, we then train with SGD and label noise for a fixed number of epochs multiple times for various values the momentum hyperparameter $\beta$. Finally, we project the weights back onto the valley before recording the final test

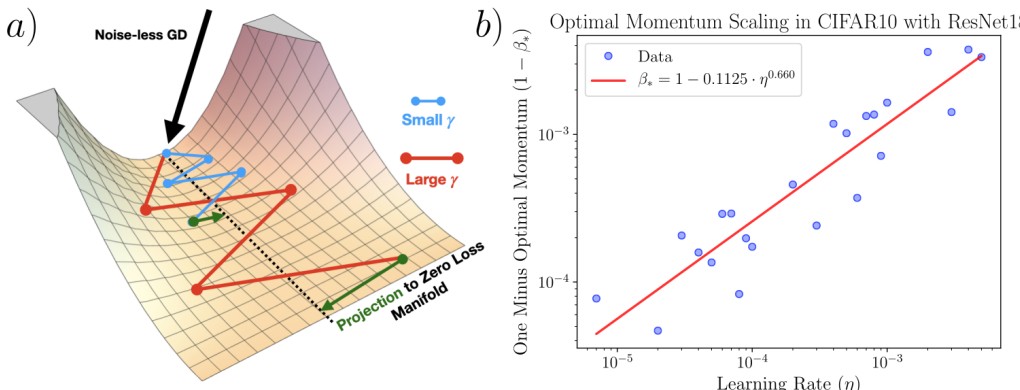

Figure 2: Classification of CIFAR10 using a ResNet18 model. Subfigure $a)$ shows our training protocol which is to first use noiseless gradient descent (black) to reach the zero-loss manifold, then to perform SGD label noise with various (blue, red) values of $\eta$ and $\beta$, before finally projecting onto the valley (green) and measuring the test accuracy. Subfigure $b)$ shows the scaling of the optimal momentum, $\beta_*$, as a function of $\eta$. We perform a power-law fit whose exponent $\gamma_* = 0.660$ matches very closely to the value implied by the theory $\gamma = \frac{2}{3}$. Notice we also extract the constant $C = 0.11$.

accuracy. This last step can be viewed as noise annealing and allows us to compare the performance of training the drift phase for the different values of $\beta$. From this procedure we extract the optimal momentum parameter $\beta_*(\eta)$ that maximizes the best test accuracy during training as a function of the learning rate, which we can then compare with the theoretical prediction.

As shown in Fig. 2(b), $1-\beta_*$ follows the power law we predicted almost exactly. The optimal choice for speedup does not have to coincide with the optimal choice for generalization. Strikingly, this optimal choice of scaling also leads to the best generalization in a realistic setting! This can be easily interpreted if we assume that the more we decrease the Hessian the better our model generalizes and by applying the fact that our scaling leads to the fastest transport along the manifold. The second important point is the value of the constant $C \approx 0.1$ found as the coefficient of the power-law fit. If we set $\eta = 1$ this corresponds to setting $\beta_* = 0.9$ which is the traditionally recommended value. The result here, can therefore be viewed as a generalization of this common wisdom. For more experiments, we refer the reader to Appendices A.2 and A.4.

## 5 CONCLUSION

We studied the implicit regularization of SGD with label noise and momentum in the limit of small noise and learning rate. We found that there is an interplay between the speedup of momentum and the limiting diffusion generated by SGD noise. This gives rise to two characteristic timescales associated to the training dynamics, and the longest timescale essentially governs the training time. Maximum acceleration is thus reached when these two timescales coincide, which lead us to identifying an optimal scaling of the hyperparameters. This optimal scaling corresponds not only to faster training but also to superior generalization. More generally, we have shown how momentum can significantly enrich the dynamics of learning with SGD, modulating between qualitatively different phases of learning characaterized by different timescales and dynamical behavior. It would be interesting to explore our scaling limit in statistical mechanical theories of learning to uncover further nontrivial effects on feature extraction and generalization in the phases of learning (Jacot et al., 2018; Roberts et al., 2021; Yang et al., 2022). For future work, it will be very interesting to generalize this result to adaptive optimization algorithms such as Adam and its variants, to use this principle to design new adaptive algorithms, and to study the interplay between the scaling we found and the hyperparameter schedule.

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

# A EXPERIMENTAL DESIGN

## A.1 UV MODEL

In our experiments with the vector UV model we aim to extract how the timescale of motion scales with $\eta$ for each $\gamma$, therefore we train the model over a sweep of both of these parameters. As a reminder, the loss of the UV model is

$$L = \frac{1}{2P} \sum_{i=1}^{P} \left( y_i - \frac{1}{\sqrt{n}} \boldsymbol{u} \cdot \boldsymbol{v} x_i \right)^2 \tag{11}$$

where $P$ is the dataset size and $\boldsymbol{u}, \boldsymbol{v}$ are $n$-dimensional vectors. We set $y_i = 0$ for simplicity, and with label noise this becomes $y_i(t) = \epsilon \cdot \xi_i(t)$ for i.i.d standard Gaussian distributed $\xi_i$. We initialize $x_i$ once (not online) as i.i.d. random standard Gaussian variables. For all experiments we choose $\epsilon = \frac{1}{2}$. We initialize with $u_i, v_i$ i.i.d. standard Gaussian distributed and keep this initialization constant over all our experiments to reduce noise associated with the specific initialization.

For each value of $\eta, \beta$ we train with label noise SGD and momentum until $|\boldsymbol{u}|^2 + |\boldsymbol{v}|^2 < \varepsilon n$ with $\varepsilon = 0.1$, thereby obtaining a time series for each of $\boldsymbol{u}(t)$ and $\boldsymbol{v}(t)$. We extract the timescale by fitting $\log \left( |\boldsymbol{u}|^2 + |\boldsymbol{v}|^2 \right)$ to a linear function and taking the slope.

## A.2 MATRIX SENSING

We also explore speedup for a well understood problem: matrix sensing. The goal is to find a low-rank matrix $X^*$ given the measurements along random matrices $A_i$: $y_i = \operatorname{Tr} A_i X^*$. Here $X^* \in \mathbb{R}^{d \times d}$ is a matrix of rank $r$ (Soudry et al., 2017; Li et al., 2018).

Blanc et al. (2020) analyze the problem of matrix sensing using SGD with label noise and show that if $X^*$ is symmetric with the hypothesis $X = UU^\top$ for some matrix $U$, then gradient descent with label noise corresponds not only to satisfying the constraints $y_i = \operatorname{Tr} A_i X$, but also to minimizing an implicit regularizer, the Frobenius norm of $U$, which eventually leads to the ground truth.

In the analogous $UV$ matrix model (with $X^*$ is an asymmetric matrix of low rank $r$), we demonstrate a considerable learning speedup by adding momentum, and show that this speedup is not monotonic with increasing $\beta$; there is a value $\beta^*$ at which the acceleration appears optimal. This non-monotonicity with an optimal $\beta^*$ is observed for both the Hessian trace and the expected test error. Assuming that in this setting we also have $\gamma = 2/3$, we can extract $C^* = (1 - \beta^*)/\eta^{2/3} \approx 0.24 \, P^{-1/3}$, which compares favorably to the upper bound we may extract from Appendix D of $\approx 0.12 \, P^{-1/3}$.

In the experiments with matrix sensing we aim to demonstrate the benefit of momentum in a popular setting. Matrix sensing corresponds to the following problem: Given a target matrix $X^* \in \mathbb{R}^{d \times d}$ of low rank $r \ll d$ and measurements $\{y_i = \operatorname{Tr} A_i X^*\}_{i=1}^{P}$ how can we reconstruct $X^*$? One way to solve this problem is to write our guess $X = UV$ the product of two other matrices, and do stochastic gradient descent on them, hoping that the implicit regularization induced by this parametrization and the learning algorithm will converge to a good low rank $X$.

### A.2.1 EXPERIMENTAL DETAILS

In our experiments we study the $d = 100, r = 5, P = 5rd = 2500$ case. We draw $(A_i)_{ij} \sim \mathcal{N}(0, 1)$ as standard Gaussians and choose $X^*$ by drawing first $(X_0)_{ij} \sim \mathcal{N}(0, 1)$ and then performing SVD and projecting onto the top $r$ singular values by zeroing out the smaller singular values in the diagonal matrix. We intitalize $U = V = I_d$. We perform SGD with momentum on the time dependent loss (with label noise depending on time)

$$L(t) = \frac{1}{dP} \sum_{i=1}^{P} \left( \epsilon \cdot \xi_i(t) + y_i - \operatorname{Tr}(A_i UV) \right)^2 \tag{12}$$

where $\epsilon^2 = 0.1, \xi_i(t) \sim \mathcal{N}(0, 1)$. We choose $\eta = 0.1$ for all of our experiments.

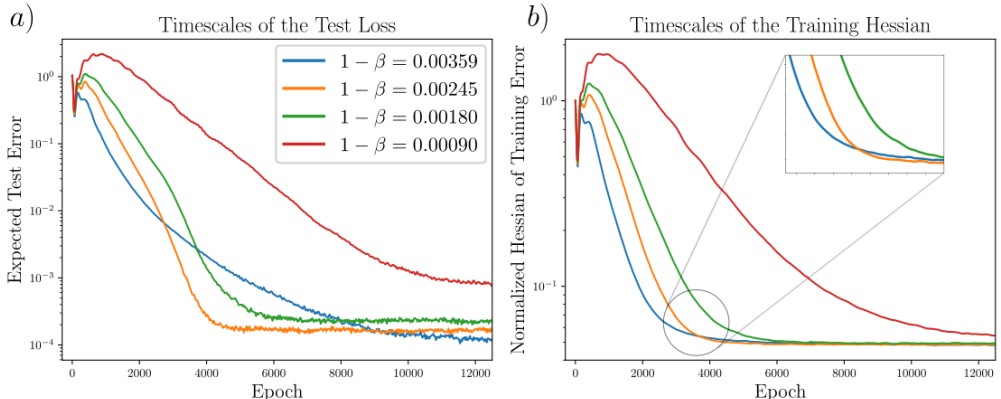

Figure 3: The expected test error, $a$), and Hessian of the training loss, $b$), in matrix sensing (with $d = 100, r = 5$, and $5rd = 2500$ samples) as a function of training epoch plotted for different values of $\beta$ at $\eta = 0.1$. The label noise variance is $0.1$. Each curve represents a different value of $\beta$. The inset shows that the orange curve crosses below the blue curve before convergence of the Hessian. Therefore, the same value of $\beta$ is optimal for both the Hessian and the expected test error — increasing or decreasing $\beta$ from this value slows down generalization.

The hessian of the loss is defined to be the Hessian averaged over the noise. Equivalently we may just set $\xi_i(t) = 0$ when we calculate the Hessian because averaging over the noise decouples the noise. Similarly when we define the expected test loss we define it as an average over all $A_i$ setting $\xi_i(t) = 0$ in order to decouple the noise. Averaging over $\xi_i(t)$ and $A_i$ would simply lead to an additional term $\langle \xi_i(t)^2 \rangle$ which would simply contribute a constant. We remove this constant for clarity. As a result, the expected test that we plot is proportional to the squared Frobenius norm of the difference between the model $UV$ and the target $X^*$,

$$\langle L \rangle = \frac{1}{d} ||UV - X^*||_F^2. \tag{13}$$

It is also interesting to note that we observe epoch-wise double descent Nakkiran et al. (2020) in this problem. In particular, we observe that the peak in the test error can be controlled by the momentum hyperparameter, and becomes especially pronounced for $\beta \to 1$.

## A.3 RESNET18 ON CIFAR10

We train our model in three steps: full batch without label noise until 100% test accuracy, SGD with label noise and momentum, and then a final projection onto the interpolating manifold. The model we use is the ResNet18 and we train on the CIFAR10 training set.

The first step is full batch gradient descent on the full CIFAR10 training set of 50,000 samples. We train with a learning rate $\eta = 0.1$ and momentum $\beta = 0.9$ and a learning rate schedule with linearly increases from 0 to $\eta$ over 600 epochs, after which it undergoes a cosine learning rate schedule for 1400 more epochs stopping the first time the network reaches 100% test accuracy which happened on epoch 1119 in our run. This model is saved and the same one is used for all future runs.

The loss function we use is cross cross entropy loss. Because we will choose a label noise level of $p = 0.2$ which corresponds to a uniformly wrong label 20% of the time, during this phase of training we train with the expected loss over this randomness. Notice that this loss is actually linear in the labels so taking the expectation is easy.

The second step involves starting from the initialization in step 1 and training with a different learning rate and momentum parameter. In this step we choose the same level of label noise $p = 0.2$ but take it to be stochastic. Additionally we use SGD instead of gradient descent with a batch size of 512. This necessitates decreasing the learning rate because noise is greatly increased as demonstrated in the main text. In this step we train for a fixed 200 epochs for any learning rate momentum

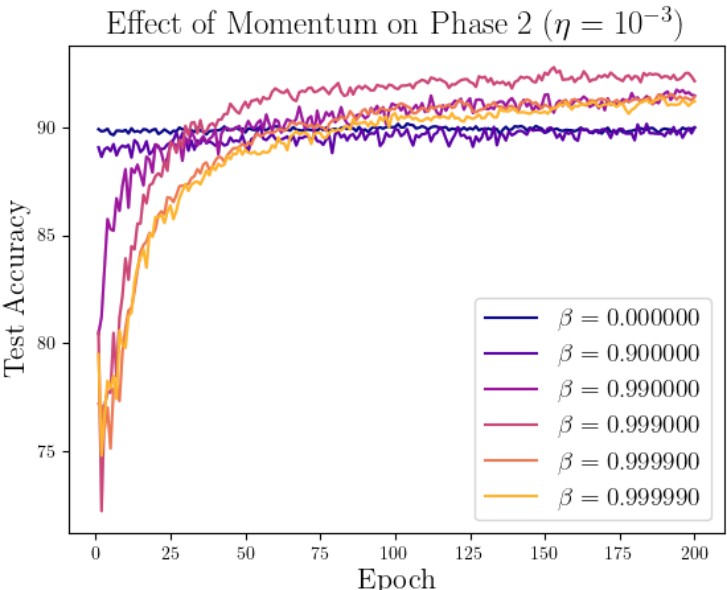

Figure 4: A sample of the curves with different momentum hyperparameters $\beta$ with $\eta = 0.001$ with Resnet on CIRAR10. The speed of increase in accuracy is non-monotonic in $\beta$: the best performance is obtained by an intermediate value of $\beta$, consistent with our predictions.

combination. We only compare runs with the same learning rate value. We show an example of the test accuracy as we train in phase 2 for $\eta = 0.001$ in figure 4.

Notice that for both too-small and too-large momentum values that the convergence to a good test accuracy value is slower. The initial transient with the decreased test accuracy happens as we start on the valley and adding noise coupled with momentum causes the weights to approach their equilibrium distribution about the valley. For wider distributions the network is farther from the optimal point on the valley. As training proceeds we see that the test accuracy actually increases over the baseline as the hessian decreases and the generalization capacity of the network increases. This happens most quickly for the momentum which matches our scaling law.

The final step is a projection onto the zero loss manifold. This step is necessary because the total width of the distribution around the zero loss manifold scales with $\frac{1}{1-\beta}$, and this will distort the results systematically at larger momentum, making them look worse than they are. We perform this projection to correct for this effect and put all momenta on an equal footing. This projection is done by training on the full batch with Adam and a learning rate of $\eta = 0.0001$ in order to accelerate training. We do not expect any significant systematic distortion by using Adam for the projection instead of gradient descent.

To determine the optimal value of $\beta$ we sweep several values of $\beta$ and observe the test accuracy after the previously described procedure. To get a more precise value of $\beta$ instead of simply selecting the one with the highest test accuracy we fit the accuracy $A(\beta)$ to

$$A(\beta) = a_{max} + \begin{cases} a_1(\beta - \beta_*) & \text{if } \beta \leq \beta_* \\ a_2(\beta - \beta_*) & \text{if } \beta \geq \beta_* \end{cases} \tag{14}$$

to the parameters $a_{max}, a_1, a_2,$ and $\beta_*$, thereby extracting $\beta_*$ for each $\eta$.

### A.4 MLP ON FASHIONMNIST

We perform a similar experiment as in section A.3 but with different model and dataset: a 6-layer MLP trained on FashionMNIST (Xiao et al., 2017) as our dataset. We perform the experiment with a 6 layer MLP with Relu activation after the first 4 layers, tanh activation after the fifth layer, and a

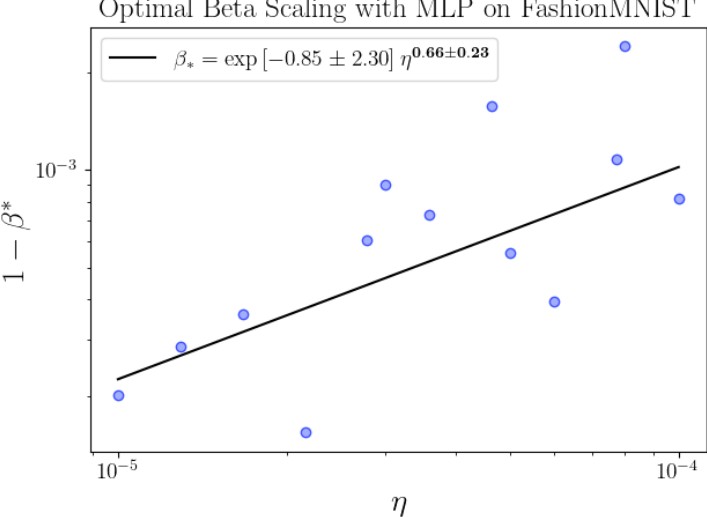

Figure 5: Scaling analysis for a 6-layer MLP on the Fashion MNIST dataset. We see that the exponent, though not very well determined, is consistant with our theoretical prediction of $\frac{2}{3}$.

linear mapping to logits. We use cross entropy loss with label noise $p = 0.2$ as before, and always start training from a reference point initialized on the zero loss manifold. This point was obtained by gradient descent on the expected loss from a random initialized point with learning rate $\eta = 0.002$ and without momentum.

After this we sweep $\eta \in [10^{-5}, 10^{-4}]$ and $\beta \in [.95, 1 - 2\eta]$ and train for 600 epochs with label noise and 400 without label noise. This allows us to obtain a test accuracy as a function of $\eta$ and $\beta$ and therefore we can obtain the best momentum hyperparameter, $\beta_*$, as a function of $\eta$ as in A.3. We extract the scaling exponent by doing a linear fit between $\log(1 - \beta_*)$ and $\log \eta$. The scaling analysis is shown in Fig. 5, and shows that the exponent is consistent with our theory.

## B    REVIEW OF RELEVANT RESULTS FROM KATZENBERGER (1991)

In this Appendix we summarize the relevant conditions and theorems from Katzenberger (1991) that we use to prove our result on the limiting drift-diffusion. We refer to Katzenberger (1991) for part of the definitions and conditions cited throughout the below. In what follows, $(\Omega^n, \mathcal{F}^n, \{\mathcal{F}_t^n\}_{t\geq 0}, P)$ will denote a filtered probability space, $Z_n$ an $\mathbb{R}^r$-valued cadlag $\{\mathcal{F}_t^n\}$-semimartingale with $Z_n(0) = 0$, $A_n$ a real-valued cadlag $\{\mathcal{F}_t^n\}$-adapted nondecreasing process with $A_n(0) = 0$, and $\tilde{\sigma} : U \to \mathbb{R}^{D\times r}$ a continuous function, where $U$ is a neighborhood of $\{\mathbf{0}\} \times \Gamma$ as defined in the main text. Also, $X_n$ is an $\mathbb{R}^D$ valued cadlag $\{\mathcal{F}_t^n\}$-semimartingale satisfying

$$X_n(t) = X_n(0) + \int_0^t \sigma_n(X_n)dZ_n + \int_0^t F(X_n)dA_n \tag{15}$$

for all $t \leq \lambda_n(K)$ and all compact $K \subset U$, where

$$\lambda_n(K) = \inf\{t \geq 0 | X_n(t-) \neq \mathring{K} \, or \, X_n(t) \neq \mathring{K}\} \tag{16}$$

be the stopping time of $X_n(t)$ to leave $\mathring{K}$, the interior of $K$. For cadlag real-valued seimimartingales $X, Y$ let $[X, Y](t)$ be defined as the limit of sums

$$\sum_{i=0}^{n-1} (X(t_{i+1}) - X(t_i))(Y(t_{i+1} - Y(t_i)) \tag{17}$$

where $0 = t_0 < t_1 < \cdots < t_n = t$ and the limit is in probability as the mesh size goes to zero. If $X$ is an $\mathbb{R}^D$-valued semimartingale, we write

$$[X] = \sum_{i=1}^{D} [X_i, X_i]. \tag{18}$$

**Condition B.1.** *For every $T > \epsilon > 0$ and compact $K \subset U$*

$$\inf_{0 \leq t \leq T \wedge \lambda_n(K) - \epsilon} (A_n(t + \epsilon) - A_n(t)) \to \infty \tag{19}$$

*as $n \to \infty$ where the infimum of the empty set is taken to be $\infty$.*

**Condition B.2.** *For every compact $K \subset U$ $\{Z_n^{\lambda_n(K)}\}$ satisfies the following: For $n \geq 1$ let $Z_n$ be a $\{\mathcal{F}_t^n\}$−semimartingale with sample paths in $D_{\mathbb{R}^d}[0, \infty)$. Assume that for some $\delta > 0$ allowing $\delta = \infty$ and every $n \geq 1$, there exist stopping times $\{\tau_n^k | k \geq 1\}$ and a decomposition of $Z_n - J_\delta(Z_n)$ into a local martingale $M_n$ plus a finite variation process $F_n$ such that $P[\tau_n^k \leq k] \leq 1/k$, $\{[M_n](t \wedge \tau_n^k) + T_{t \wedge \tau_n^k}(F_n) | n \geq 1\}$ is uniformly integrable for every $t \geq 0$ and $k \geq 1$ and*

$$\lim_{\gamma \to 0} \limsup_{n \to \infty} P \left[ \sup_{0 \leq t \leq T} (T_{t+\gamma}(F_n) - T_t(F_n)) > \epsilon \right] = 0 \tag{20}$$

*for every $\epsilon > 0$ and $T > 0$. Also as $n \to \infty$ and for any $T > 0$*

$$\sup_{0 < t \leq T \wedge \lambda_n(K)} |\Delta Z_n(t)| \to 0 \tag{21}$$

**Condition B.3.** *The process*

$$\bar{Z}_n(t) = \sum_{0 < s \leq t} \Delta Z_n(s) \Delta A_n(s) \tag{22}$$

*exists, is an $\{\mathcal{F}_t^n\}$−semimartingale, and for every compact $K \subset U$, the sequence $\{\bar{Z}_n^{\lambda_n(K)}\}$ is relatively compact and satisfies Condition 4.1 in* Katzenberger *(1991).*

**Theorem B.4** (Theorem 7.3 in Katzenberger (1991))**.** *Assume that $\Gamma$ is $C^2$ and for every $y \in \Gamma$, the matrix $\partial F(y)$ has $D - M$ eigenvalues in $D(1)$. Assume (B.1),(B.2) and (B.3) hold, $\Phi$ is $C^2$ (or $F$ is $LC^2$) and $X_n(0) \Rightarrow X(0) \in U$. Let*

$$Y_n(t) = X_n(t) - \psi(X(0), A(t)) + \Phi(X(0)) \tag{23}$$

*and, for a compact $K \subset U$, let*

$$\mu_n(K) = \inf\{t \geq 0 | Y_n(t-) \notin \mathring{K} \text{ or } Y_n(t) \notin \mathring{K}\}. \tag{24}$$

*Then for every compact $K \subset U$, the sequence $\{(Y_n^{\mu_n(K)}, Z_n^{\mu_n(K)}, \mu_n(K))\}$ is relatively compact in $D_{\mathbb{R}^{2D \times r}}[0, \infty) \times [0, \infty]$ (see* Katzenberger *(1991) for details about the topology). If $(Y, Z, \mu)$ is a limit of this sequence then $(Y, Z)$ is a continuous semimartingale, $Y(t) \in \Gamma$ for every $t$ almost surely, $\mu \geq \inf\{t \geq 0 | Y(t) \notin \mathring{K}\}$ almost surely, and*

$$Y(t) = Y(0) + \int_0^{t \wedge \mu} \partial \Phi(Y) \tilde{\sigma}(Y) dZ + \frac{1}{2} \sum_{ijkl} \int_0^{t \wedge \mu} \partial_{ij} \Phi(Y) \tilde{\sigma}^{ik}(Y) \tilde{\sigma}^{jl}(Y) d[Z^k, Z^l]. \tag{25}$$

## B.1 APPLYING THEOREM B.4

Recall the equations of motion of stochastic gradient descent

$$\pi_{k+1}^{(n)} = \beta \pi_k^{(n)} - \nabla L(w_k^{(n)}) + \epsilon_n \sigma(w_k^{(n)}) \xi_k, \qquad w_{k+1}^{(n)} = w_k^{(n)} + \eta \pi_{k+1}^{(n)}, \tag{26}$$

where $\sigma(w) \in \mathbb{R}^{D \times r}$ is the noise function evaluated at $w \in \mathbb{R}^D$, and $\xi_k \in \mathbb{R}^r$ is a noise vector drawn i.i.d. at every timestep $k$ with zero mean and unit variance. We now show that this equation satisfies all the properties required by Theorem B.4.

The manifold $\Gamma$ is the fixed point manifold of (non-stochastic) gradient descent. $\{\mathbf{0}\} \times \Gamma$ is a $\mathcal{C}^2$ manifold because $\Gamma$ is $\mathcal{C}^2$, which follows from assumption 3.1. The flow $F(w, \pi) =$

$(\eta(\beta\pi - \nabla L(w)), \beta\pi - \nabla L(w))$. As shown in Appendix E, $dF$ has exactly $M$ zero eigenvalues on $\Gamma \cap K$. $F$ inherits the differentiable and locally Lipschitz properties from $\nabla L$, and therefore satisfies the conditions of B.4.

Next, notice that the noise function $\tilde{\sigma} : R^{2D} \to \mathbb{R}^{2D \times r}$ is continuous because $\sigma$ is.

Now we define $A_n$ and $Z_n$ (as in the main text) so that $X_n$ reproduces the dynamics in equation (26), except with a new time parameter $t = k\epsilon_n^2$.

$$A_n(t) = \left\lfloor \frac{t}{\epsilon_n^2} \right\rfloor, \qquad Z_n(t) = \epsilon_n \sum_{k=1}^{A_n(t)} \xi_k \tag{27}$$

So that, with these choices, Eq. (15) precisely corresponds to (26), up to the rescaling $t = k\epsilon_n^2$.

Now we show that $A_n, Z_n$ satisfy the conditions of B.4. Clearly $A_n(0) = Z_n(0)$ by definition. Then

$$A_n(t + \varepsilon) - A_n(t) = \left\lfloor \frac{t + \varepsilon}{\epsilon_n^2} \right\rfloor - \left\lfloor \frac{t}{\epsilon_n^2} \right\rfloor \geq \frac{t + \varepsilon - t}{\epsilon_n^2} - 2 = \frac{\varepsilon}{\epsilon_n^2} - 2 \to \infty \tag{28}$$

when we take $\epsilon_n \to 0$, thus recovering condition B.1.

By definition $Z_n$ is a martingale. Notice also by the definition of $Z_n$, because $\xi_k$ is i.i.d. with variance 1, that $Z_n(t)$ has variance $A_n(t)\epsilon_n^2 \leq t$ which is uniformly bounded and hence $Z_n(t)$ is uniformly integrable for stopping times $\tau_n^k = 2k > k$. Also note that $\left|\Delta Z_n(k\epsilon_n^2)\right| = |\epsilon_n \xi_k|$ which goes to zero in probability as $\epsilon$ becomes small because $\xi_k$ has bounded variance, and $\Delta Z_n(t)$ is zero otherwise. This shows that $Z_n$ satisfies condition B.2. Because $Z_n$ and $A_n$ are discontinuous at the same time we automatically satisfy condition B.3 as pointed out by Katzenberger.

This shows that we satisfy the conditions of Theorem B.4, therefore we have the following

**Lemma B.5.** *The SGD equations formulated as in (26) satisfy all the conditions of Theorem B.4.*

## B.2 EQUIVALENCE BETWEEN EQ. (1) AND EQ. (4)

Eq. (4) is the rewriting of eq. (1) in the form presented in Katzenberger (1991), on which our theory is based. We show the equivalence below, and include the definitions of $A$ and $Z$ to keep this section self-contained. The statement is that Eq. (1), i.e.

$$\pi_{k+1} = \beta\pi_k - \nabla L(w_k) + \epsilon\sigma(w_k)\xi_k, \qquad w_{k+1} = w_k + \eta\pi_{k+1}, \tag{29}$$

can be rewritten in the form of eq. (4)

$$X_n(t) = X_n(0) + \int_0^t \tilde{\sigma}(X_n)dZ_n + \int_0^t F(X_n)dA_n, \tag{30}$$

$$A_n(t) = \left\lfloor \frac{t}{\epsilon_n^2} \right\rfloor, \qquad Z_n(t) = \epsilon_n \sum_{k=1}^{A_n(t)} \xi_k \tag{31}$$

where $X_n(t) = (\pi_k, w_k)$, with the correspondence $t = k\epsilon_n^2$, where $k$ denotes the SGD time step. Also, $\tilde{\sigma}(X) = (\sigma, \eta\sigma)$ and $F(X) = ((\beta - 1)\pi - \nabla L(w), \eta(\beta\pi - \nabla L(w)))$.

Before we begin the proof, we will need the following fact (see Sec. 2 of Katzenberger (1991)):

$$\int_s^t f dg = \int_s^t f dg^c + \sum_{s < r \leq t} f(r-)\Delta g(r), \tag{32}$$

where $f$ and $g$ are càdlàg functions, in particular they are right-continuous and have left limits everywhere. The integral above is done with respect to the measure $dg$, the differential of a function $g$. The sum is taken over all $r \in (s, t]$ where $g$ is discontinuous and the notation $\Delta g(r) = g(r) - g(r-)$ indicates the discontinuity of $g$ at $r$, where $g(r-) = \lim_{u \to r^-} g(u)$ indicates the left limit of $g$ at $r$. Finally $g^c$ denotes the continuous part of $g$

$$g^c(t) = g(t) - \sum_{0 < s \leq t} \Delta g(s), \qquad t \geq 0. \tag{33}$$

We now show by induction that $X_n(t = k\epsilon_n^2)$ solves the first equation above. Note that

$$dA_n(t) = \sum_{k=-\infty}^{\infty} \delta(t - \epsilon_n^2 k), \qquad dZ_n(t) = \epsilon_n \sum_{k=\infty}^{\infty} \xi_k \delta(t - \epsilon_n^2 k). \tag{34}$$

For brevity we will drop the subscript $n$ and let $k\epsilon^2 = t$. Consider

$$X(t + \epsilon^2) = \int_0^{t+\epsilon^2} \tilde{\sigma}(X(s))dZ(s) + \int_0^{t+\epsilon^2} F(X(s))dA(s) \tag{35}$$

$$= X(t) + \int_t^{t+\epsilon^2} \tilde{\sigma}(X(s))dZ(s) + \int_t^{t+\epsilon^2} F(X(s))dA(s) \tag{36}$$

$$= X(t) + \sum_{t<s\leq t+\epsilon^2} \tilde{\sigma}(X(s-))\Delta Z(s) + \sum_{t<s\leq t+\epsilon^2} F(X(s-))\Delta A(s) \tag{37}$$

where in the last step we use eq. (32) as $A, Z, \tilde{\sigma}$ and $F$ are càdlàg, and that $dA^c = dZ^c = 0$. The sums are taken over all $s \in (k\epsilon^2, (k+1)\epsilon^2]$ where $Z(s)$ and $A(s)$ are discontinuous. The only point of discontinuity of $A$ and $Z$ in this interval is at $s = t + \epsilon^2$, so

$$X(t + \epsilon^2) = X(t) + \epsilon\tilde{\sigma}(X((t + \epsilon^2)-))\xi_{k+1} + F(X((t + \epsilon^2)-)) \tag{38}$$

because the jumps of $Z$ and $A$ at $s = t + \epsilon^2$ are $\epsilon\xi_{k+1}$ and 1, respectively. Now we must determine the left limit of $X$ at $t + \epsilon^2$. Notice that for $0 < \delta < \epsilon$

$$X(t + \delta^2) = X(t) + \sum_{t<s\leq t+\delta^2} \tilde{\sigma}(X(s-))\Delta Z(s) + \sum_{t<s\leq t+\delta^2} F(X(s-))\Delta A(s) \tag{39}$$

$$= X(t) \tag{40}$$

because $A$ and $Z$ are continuous on $(t, t+\delta^2]$. Hence the left limit of $X((t+\epsilon^2)-) = X(t)$. Putting these equations together we find that

$$X((k+1)\epsilon^2) = X(t + \epsilon^2) = X(k\epsilon^2) + \epsilon\tilde{\sigma}(X(k\epsilon^2))\xi_{k+1} + F(X(k\epsilon^2)) \tag{41}$$

which, using the definition of $X(k\epsilon^2) = (\pi_k, w_k)$, is eq. (29), thus proving equivalence.

## C    EXPLICIT EXPRESSION OF LIMITING DIFFUSION IN MOMENTUM SGD

In this section we provide the proof of Theorem 3.4. Recall that $\Phi$ satisfies

$$\Phi(x + F(x)) = \Phi(x), \tag{42}$$

where $x = (\pi, w)$, and $F$ given in (132). To obtain the explicit expression of the limiting drift-diffusion, according to Theorem (B.4), and keeping into account Assumption (3.2), we need to determine $\Phi$ up to its second derivatives. To this aim, we shall expand Eq. (42) up to second order in the series expansion in $F$. In components, this reads:

$$\eta\partial_{w_i}\Phi^j(\pi^i - g^i) + \partial_{\pi_i}\Phi^j(-C\eta^\gamma\pi^i - g^i) + \frac{1}{2}\eta^2\partial_{w_i}\partial_{w_k}\Phi^j(\pi^i - g^i)(\pi^k - g^k)$$

$$+\frac{1}{2}\eta\partial_{w_i}\partial_{\pi_k}\Phi^j(\pi^i - g^i)(-C\eta^\gamma\pi^k - g^k) + \frac{1}{2}\eta\partial_{\pi_i}\partial_{w_k}\Phi^j(-C\eta^\gamma\pi^i - g^i)(\pi^k - g^k) \tag{43}$$

$$+\frac{1}{2}\partial_{\pi_i}\partial_{\pi_k}\Phi^j(-C\eta^\gamma\pi^i - g^i)(-C\eta^\gamma\pi^k - g^k) = 0$$

subject to the boundary condition

$$\Phi(\pi, w)|_{w\in\Gamma, \pi=0} = w. \tag{44}$$

Here, $g^i = \partial_i L$. In Eq. (43), we already substituted $\beta = 1 - C\eta^\gamma$. In what follows, we shall find $\Phi$ to leading order in $\eta$ as $\eta \to 0$. We will solve the above problem by performing a series expansion in $\Phi$ around a point $\bar{w} \in \Gamma$ and $\pi = 0$ up to second order:

$$\Phi(\pi, \bar{w} + \delta w) = \Phi_{00} + \Phi_{01}^i \delta w^i + \Phi_{10}^i \pi^i + \frac{1}{2}\Phi_{02}^{ij}\delta w^i\delta w^j + \Phi_{11}^{ij}\pi^i\delta w^j + \frac{1}{2}\Phi_{20}^{ij}\pi^i\pi^j + \cdots. \tag{45}$$

For example, we have

$$\Phi_{01}^i = \frac{\partial \Phi}{\partial w^i}, \qquad \Phi_{11}^{ij} = \frac{\partial^2 \Phi}{\partial \pi^i \partial w^j}, \qquad \Phi_{20}^{ij} = \frac{\partial^2 \Phi}{\partial \pi^i \partial \pi^j}. \tag{46}$$

More precisely, we regard this as an expansion in powers of $\delta w$ and $\pi$: $\Phi_{00}$ is zeroth order, $\Phi_{10}, \Phi_{01}$ are first order, and the remaining terms are second order. We will occasionally write explicitly the index of $\Phi$, e.g. $\Phi_{02}^{k,ij} = \frac{\partial \Phi^k}{\partial w^i \partial w^j}$. It will be useful to introduce the longitudinal projector onto $\Gamma$, $P_L(w) : \mathbb{R}^D \to T_w\Gamma$: $P_L v = v$. The transverse projector is then $P_T = \mathrm{Id} - P_L$. We will also decompose various tensors using these projectors, e.g.

$$\Phi_{11}^{ij} = \Phi_{11LL}^{ij} + \Phi_{11LT}^{ij} + \Phi_{11TL}^{ij} + \Phi_{11TT}^{ij}, \tag{47}$$

with $\Phi_{11LT}^{ij} = \Phi_{11}^{kl} P_L^{ik} P_T^{jl}$. Note that the Hessian $H = \nabla g \in \mathbb{R}^{D \times D}$ satisfies $HH^\dagger = P_T$, where $H^\dagger$ denotes the pseudoinverse.

At zeroth order, we obviously have $\Phi_{00}(w) = w$.

**Lemma C.1.** *The first order terms in the series expansion (45) are given by*

$$\Phi_{01T} = \Phi_{10T} = 0, \qquad \Phi_{01L} = P_L, \qquad \Phi_{10L} = C^{-1}\eta^{1-\gamma}P_L. \tag{48}$$

*Proof.* Suppose $\hat{w}(s) \in \mathbb{R}^D$ is a curve lying on $\Gamma$. Then due to the boundary condition (44), $\partial_s\Phi(\pi = 0, \hat{w}) = \Phi_{01}^i \partial_s \hat{w}^i = \partial_s \hat{w}^i$. This means that $\Phi_{01L} = P_L$. Now from (43),

$$\eta\Phi_{01}^i(\pi^i - \partial_j g^i \delta w^j) + \Phi_{10}^i(-C\eta^\gamma \pi^i - \partial_j g^i \delta w^j) = 0. \tag{49}$$

This condition should hold for any $\pi$ and $\delta w$, therefore we arrive at

$$\Phi_{10} = C^{-1}\eta^{1-\gamma}\Phi_{01} \tag{50}$$

and

$$(\eta\Phi_{01}^i + \Phi_{10}^i)\partial_k g^i = 0. \tag{51}$$

Decomposing into longitudinal and transverse components, and noting that the Hessian satisfies $P_L H = 0$, the above equation becomes

$$\eta\Phi_{01T}^i + \Phi_{10T}^i = 0, \tag{52}$$

which together with (50) and the above discussion gives

$$\Phi_{01T} = \Phi_{10T} = 0, \qquad \Phi_{01L} = P_L, \qquad \Phi_{10L} = C^{-1}\eta^{1-\gamma}P_L, \tag{53}$$

which concludes the first order analysis. $\qquad\square$

We now need a Lemma, which requires Definition (3.3) in the main text. We report it here for convenience:

**Definition C.2.** For a symmetric matrix $H \in \mathbb{R}^D \times \mathbb{R}^D$, and $W_H = \{\Sigma \in \mathbb{R}^D \times \mathbb{R}^D : \Sigma = \Sigma^\top, HH^\dagger\Sigma = H^\dagger H\Sigma = \sigma\}$, we define the operator $\tilde{\mathcal{L}}_H : W_H \to W_H$ with $\tilde{\mathcal{L}}_H S \equiv \{H, S\} + \frac{1}{2}C^{-2}\eta^{1-2\gamma}[[S,H],H]$, with $[S,H] = SH - HS$.

**Lemma C.3.** *The inverse of the operator $\tilde{\mathcal{L}}_H$ is unique.*

*Proof.* Let us go to a basis where $H$ is diagonal, with eigenvalues $\lambda_i$. In components, the equation $\tilde{\mathcal{L}}_H S = M$ reads

$$(\lambda_i + \lambda_j)S_{ij} + \frac{1}{2}C^{-2}\eta^{1-2\gamma}(\lambda_i - \lambda_j)^2 S_{ij} + C^{-1}\eta^{1-\gamma}M_{ij} = 0 \tag{54}$$

which has a unique solution, with

$$S_{ij} = -C^{-1}\eta^{1-\gamma}\left(\lambda_i + \lambda_j + \frac{1}{2}C^{-2}\eta^{1-2\gamma}(\lambda_i - \lambda_j)^2\right)^{-1} M_{ij}. \tag{55}$$

$\square$

**Lemma C.4.** *The second order terms in the series expansion ([45](#)) are given by*

$$\Phi_{02LL}^{j,ik} = -(H^\dagger)^{jl}\partial_i^L H^{ln}P_L^{nk}, \quad \Phi_{02LT}^{j,ik} = -P_L^{jl}\partial_i^L H^{ln}(H^\dagger)^{nk}, \quad \Phi_{02TT}^{j,ik} = O\left(\eta^{min\{0,1-2\gamma\}}\right)$$

$$\tag{56}$$

$$\Phi_{11LL}^{j,ik} = -C^{-1}\eta^{1-\gamma}(H^\dagger)^{jl}\partial_i^L H^{ln}P_L^{nk}, \quad \Phi_{11TL}^{j,ik} = -C^{-1}\eta^{1-\gamma}P_L^{jl}\partial_k^L H^{ln}(H^\dagger)^{ni} \tag{57}$$

$$\Phi_{11TT}^{j,ik} = -C^{-1}\eta^{1-\gamma}\tilde{\mathcal{L}}_H^{-1}(M^{(j)})_{ik} - \frac{1}{2}C^{-3}\eta^{2-3\gamma}[H,\tilde{\mathcal{L}}_H^{-1}M^{(j)}]_{ik} \tag{58}$$

$$\Phi_{20LL}^{j,ik} = -\frac{1}{2}C^{-2}\eta^{2-2\gamma}\left((H^\dagger)^{jl}\partial_i^L H^{ln}P_L^{nk} + (H^\dagger)^{jl}\partial_k^L H^{ln}P_L^{ni}\right) \tag{59}$$

$$\Phi_{20TL}^{j,ik} = -\frac{1}{2}C^{-2}\eta^{2-2\gamma}\left(P_L^{jl}\partial_k^L H^{ln}(H^\dagger)^{ni} + P_L^{jl}\partial_i^L H^{ln}(H^\dagger)^{nk}\right) \tag{60}$$

$$\Phi_{20TT}^{j,ik} = -C^{-2}\eta^{2-2\gamma}\tilde{\mathcal{L}}_H^{-1}(P_T\partial_j^L H P_T)_{ik} \tag{61}$$

*where*

$$(M^{(j)})_{kl} = P_T^{ki}\partial_j^L H_{ni}P_T^{nl} \tag{62}$$

*Proof.* Consider a path $\hat{w}(s)$ lying on $\Gamma$. From $P_T H = H$ we have

$$\frac{dP_T}{ds}H = (\text{Id} - P_T)\frac{dH}{ds} = P_L\frac{dH}{ds}, \tag{63}$$

and thus, using $HH^\dagger = P_T$, we find

$$\frac{dP_T}{ds}P_T = P_L\frac{dH}{ds}H^\dagger, \qquad P_T\frac{dP_T}{ds} = H^\dagger\frac{dH}{ds}P_L, \tag{64}$$

where the second equation is obtained from the first by taking the transpose. Putting the last two relations together, we find

$$\frac{dP_L}{ds} = -\frac{dP_T}{ds} = -P_T\frac{dP_T}{ds} - \frac{dP_T}{ds}P_T = -H^\dagger\frac{dH}{ds}P_L - P_L\frac{dH}{ds}H^\dagger. \tag{65}$$

From ([48](#)) we can then write

$$\partial_i^L \Phi_{01}^{j,k} = -(H^\dagger)^{jl}\partial_i^L H^{ln}P_L^{nk} - P_L^{jl}\partial_i^L H^{ln}(H^\dagger)^{nk}, \tag{66}$$

where $\partial_i^L = P_L^{ij}\partial_j$. This leads to

$$\Phi_{02LL}^{j,ik} = P_L^{kl}\partial_i^L\Phi_{01}^{j,l} = -(H^\dagger)^{jl}\partial_i^L H^{ln}P_L^{nk} \tag{67}$$

$$\Phi_{02LT}^{j,ik} = P_T^{kl}\partial_i^L\Phi_{01}^{j,l} = -P_L^{jl}\partial_i^L H^{ln}(H^\dagger)^{nk}. \tag{68}$$

Also note that $\Phi_{02LT}^{j,ik} = \Phi_{02TL}^{j,ki}$, so the only component still to be determined in $\Phi_{02}$ is $\Phi_{02TT}$.

The next step is to expand Eq. ([43](#)) to second order:

$$\pi^i\pi^k\left(\eta(1+C\eta^\gamma)\Phi_{11}^{ki} - C\eta^\gamma(1-\tfrac{1}{2}C\eta^\gamma)\Phi_{20}^{ik} + \tfrac{1}{2}\eta^2\Phi_{02}^{ik}\right)$$
$$+\delta w^k\pi^i\left(\eta\Phi_{02}^{ik} - \eta^2 H_{kj}\Phi_{02}^{ji} - \eta\Phi_{11}^{il}H_{kl} - C\eta^\gamma\Phi_{11}^{ik} - \eta(1-C\eta^\gamma)\Phi_{11}^{(ij)}H_{kj} - (1-C\eta^\gamma)\Phi_{20}^{li}H_{kl}\right)$$
$$-\delta w^k\delta w^l\left(\eta\Phi_{02}^{ik}H_{li} + \tfrac{1}{2}\eta^2\Phi_{02}^{ji}H_{lj}H_{ki} + \Phi_{11}^{ik}H_{li} + \eta\Phi_{11}^{ji}H_{kj}H_{li}\right.$$
$$\left. +\tfrac{1}{2}(\eta\Phi_{01}^i + \Phi_{10}^i)\partial_k H_{li} + \tfrac{1}{2}\Phi_{20}^{ji}H_{lj}H_{ki}\right) = 0, \tag{69}$$

where $A^{(ij)} = \frac{1}{2}(A^{ij} + A^{ji})$ denotes the symmetric part. Neglecting various terms that are subleading at small $\eta$, gives

$$\pi^i\pi^k\left(\eta\Phi_{11}^{ki} - C\eta^\gamma\Phi_{20}^{ik} + \tfrac{1}{2}\eta^2\Phi_{02}^{ik}\right)$$
$$+\delta w^k\pi^i\left(\eta\Phi_{02}^{ik} - C\eta^\gamma\Phi_{11}^{ik} - \Phi_{20}^{li}H_{kl}\right)$$
$$-\delta w^k\delta w^l\left(\eta\Phi_{02}^{ik}H_{li} + \Phi_{11}^{ik}H_{li} + \tfrac{1}{2}(\eta\Phi_{01}^i + \Phi_{10}^i)\partial_k H_{li} + \tfrac{1}{2}\Phi_{20}^{ji}H_{lj}H_{ki}\right) = 0, \tag{70}$$

The first line immediately gives

$$\Phi_{20}^{ij} = C^{-1}\eta^{1-\gamma}\Phi_{11}^{(ij)} + \frac{1}{2}C^{-1}\eta^{2-\gamma}\Phi_{02}^{ij}\,. \tag{71}$$

Taking the second line of (70) and projecting onto the longitudinal part the index $k$ gives

$$\eta\beta\Phi_{02LL}^{ik} - C\eta^{\gamma}\Phi_{11LL}^{ik} = 0 \tag{72}$$

$$\eta\beta\Phi_{02TL}^{ik} - C\eta^{\gamma}\Phi_{11TL}^{ik} = 0 \tag{73}$$

and using (67),(68),

$$\Phi_{11LL}^{j,ik} = C^{-1}\eta^{1-\gamma}\Phi_{02LL}^{j,ik} = -C^{-1}\eta^{1-\gamma}(H^{\dagger})^{jl}\partial_i^L H^{ln}P_L^{nk} \tag{74}$$

$$\Phi_{11TL}^{j,ik} = C^{-1}\eta^{1-\gamma}\Phi_{02TL}^{j,ik} = -C^{-1}\eta^{1-\gamma}P_L^{jl}\partial_k^L H^{ln}(H^{\dagger})^{ni} \tag{75}$$

Projecting the second line of (70) on the transverse part of the index $k$ we have

$$\eta\Phi_{02LT}^{j,ik} - C\eta^{\gamma}\Phi_{11LT}^{j,ik} - \Phi_{20TL}^{j,li}H_{kl} = 0 \tag{76}$$

$$\eta\Phi_{02TT}^{j,ik} - C\eta^{\gamma}\Phi_{11TT}^{j,ik} - \Phi_{20TT}^{j,li}H_{kl} = 0 \tag{77}$$

Using (71) and keeping (68) into account, Eq. (76) becomes, neglecting subleading terms in $\eta$,

$$-\eta P_L^{jl}\partial_i^L H^{ln}(H^{\dagger})^{nk} - C\eta^{\gamma}\Phi_{11LT}^{ik} - \frac{1}{2}C^{-1}\eta^{1-\gamma}\Phi_{11TL}^{li}H_{kl} - \frac{1}{2}C^{-1}\eta^{1-\gamma}\Phi_{11LT}^{il}H_{kl} = 0\,. \tag{78}$$

Using (75) in (78),

$$-\eta P_L^{jl}\partial_i^L H^{ln}(H^{\dagger})^{nk} - C\eta^{\gamma}\Phi_{11LT}^{ik}$$
$$+\frac{1}{2}C^{-2}\eta^{2-2\gamma}P_L^{jp}\partial_i^L H^{pn}(H^{\dagger})^{nl}H_{kl} - \frac{1}{2}C^{-1}\eta^{1-\gamma}\Phi_{11LT}^{il}H_{kl} = 0\,, \tag{79}$$

and simplifying,

$$-\eta P_L^{jl}\partial_i^L H^{ln}(H^{\dagger})^{nk} - C\eta^{\gamma}\Phi_{11LT}^{ik}$$
$$+\frac{1}{2}C^{-2}\eta^{2-2\gamma}P_L^{jl}\partial_i^L H^{ln}P_T^{nk} - \frac{1}{2}C^{-1}\eta^{1-\gamma}\Phi_{11LT}^{il}H_{kl} = 0\,. \tag{80}$$

which determines $\Phi_{11LT}$ in close form. Indeed, the above has the form, in matrix notation

$$-\frac{1}{2}C^{-1}\eta^{1-\gamma}\Phi_{11LT}H - C\eta^{\gamma}\Phi_{11LT} = M\,, \tag{81}$$

and can be immediately inverted to solve for $\Phi_{11LT}$. The only two undetermined components now are $\Phi_{11TT}$ and $\Phi_{02TT}$. One condition is obtained from (77) which gives, keeping (71) into account,

$$\eta\Phi_{02TT}^{j,ik} - C\eta^{\gamma}\Phi_{11TT}^{j,ik} - C^{-1}\eta^{1-\gamma}\Phi_{11TT}^{j,(li)}H_{kl} = 0\,, \tag{82}$$

and thus

$$\Phi_{02TT}^{j,ik} = C\eta^{\gamma-1}\Phi_{11TT}^{j,ik} + C^{-1}\eta^{-\gamma}\Phi_{11TT}^{j,(ni)}H_{kn}\,. \tag{83}$$

Further taking symmetric and antisymmetric part in $ik$ of the above gives

$$\Phi_{02TT}^{j,ik} = C\eta^{\gamma-1}\Phi_{11TT}^{j,(ik)} + \frac{1}{2}C^{-1}\eta^{-\gamma}\Phi_{11TT}^{j,(ni)}H_{kn} + \frac{1}{2}C^{-1}\eta^{-\gamma}\Phi_{11TT}^{j,(nk)}H_{in} \tag{84}$$

$$0 = C\eta^{\gamma-1}\Phi_{11TT}^{j,[ik]} + \frac{1}{2}C^{-1}\eta^{-\gamma}\Phi_{11TT}^{j,(ni)}H_{kn} - \frac{1}{2}C^{-1}\eta^{-\gamma}\Phi_{11TT}^{j,(nk)}H_{in}\,. \tag{85}$$

The other condition comes from the third line of (70) which, using (48) and (71), and neglecting subleading terms in $\eta$, gives

$$\eta\Phi_{02}^{j,ik}H_{li} + \eta\Phi_{02}^{j,il}H_{ki} + \Phi_{11}^{j,ik}H_{li} + \Phi_{11}^{j,il}H_{ki} + C^{-1}\eta^{1-\gamma}P_L^{ji}\partial_k H_{li} = 0 \tag{86}$$

Projecting the $k$ and $l$ indices on the longitudinal part, gives $P_L^{ji}\partial_k^L H_{li}P_L^{ln} = 0$, which is an identity that can be checked from (63). Projecting $k$ on the longitudinal part and $l$ on the transverse part gives

$$\eta\Phi_{02TL}^{j,ik}H_{li} + \Phi_{11TL}^{j,ik}H_{li} + C^{-1}\eta^{1-\gamma}P_L^{ji}\partial_k^L H_{ni}P_T^{nl} = 0\,, \tag{87}$$

which is implied by (75), indeed

$$
\begin{aligned}
(\eta\Phi^{j,ik}_{02TL} + \Phi^{j,ik}_{11TL})H_{li} &= C^{-1}\eta^{1-\gamma}\Phi^{j,ik}_{02TL}H_{li} \\
&= -C^{-1}\eta^{1-\gamma}P^{jl}_L\partial^L_k H^{ln}(H^\dagger)^{ni}H_{li} = -C^{-1}\eta^{1-\gamma}P^{jl}_L\partial^L_k H^{ln}P^{nl}_T,
\end{aligned}
\tag{88}
$$

therefore (87) does not give a new condition. The only new condition comes from projecting the $k$ and $l$ indices of (86) on the transverse direction, giving

$$
\eta\Phi^{j,ik}_{02TT}H_{li} + \eta\Phi^{j,il}_{02TT}H_{ki} + \Phi^{j,ik}_{11TT}H_{li} + \Phi^{j,il}_{11TT}H_{ki} + C^{-1}\eta^{1-\gamma}P^{ji}_L\partial^T_k H_{ni}P^{nl}_T = 0\,.
\tag{89}
$$

Plugging in (83),

$$
\begin{aligned}
&C\eta^\gamma\Phi^{j,ik}_{11TT}H_{li} + C^{-1}\eta^{1-\gamma}\Phi^{j,(in)}_{11TT}H_{kn}H_{li} + C\eta^\gamma\Phi^{j,il}_{11TT}H_{ki} + C^{-1}\eta^{1-\gamma}\Phi^{j,(in)}_{11TT}H_{ln}H_{ki} \\
&+\Phi^{j,ik}_{11TT}H_{li} + \Phi^{j,il}_{11TT}H_{ki} + C^{-1}\eta^{1-\gamma}P^{ji}_L\partial^T_k H_{ni}P^{nl}_T = 0\,.
\end{aligned}
\tag{90}
$$

Neglecting subleading terms in $\eta$ this can be rewritten as

$$
2C^{-1}\eta^{1-\gamma}\Phi^{j,(in)}_{11TT}H_{kn}H_{li} + \Phi^{j,ik}_{11TT}H_{li} + \Phi^{j,il}_{11TT}H_{ki} + C^{-1}\eta^{1-\gamma}\partial^L_j H_{ni}P^{ki}_T P^{nl}_T = 0\,.
\tag{91}
$$

where we recall that the last term is symmetric in $k$ and $l$. In matrix notation this reads

$$
C^{-1}\eta^{1-\gamma}H(\Phi^j_{11TT} + \Phi^{jt}_{11TT})H + H\Phi^j_{11TT} + \Phi^{jt}_{11TT}H + C^{-1}\eta^{1-\gamma}M^{(j)} = 0\,,
\tag{92}
$$

where

$$
(M^{(j)})_{kl} = P^{ki}_T\partial^L_j H_{ni}P^{nl}_T
\tag{93}
$$

is the longitudinal derivative of the Hessian, projected on the transverse directions. Decomposing $\Phi^j_{11TT}$ into symmetric and antisymmetric parts $\Phi^j_{11TT} = S^{(j)} + A^{(j)}$, Eq. (92) reads (suppressing the index $j$)

$$
2C^{-1}\eta^{1-\gamma}HSH + HS + SH + HA - AH + C^{-1}\eta^{1-\gamma}M = 0\,.
\tag{94}
$$

The first term is subleading in $\eta$, therefore we have

$$
HS + SH + HA - AH + C^{-1}\eta^{1-\gamma}M = 0\,.
\tag{95}
$$

This equation together with (85), which in matrix form reads

$$
C^{-1}\eta^{-\gamma}(SH - HS) + 2C\eta^{\gamma-1}A = 0\,,
\tag{96}
$$

determine $S$ and $A$, and thus $\Phi_{11TT}$. Note that $M^j$ has only a longitudinal component in the index $j$, therefore the transverse parts of $S$ and $A$ vanish, i.e.

$$
P^{pj}_T\Phi^{j,ik}_{11TT} = 0\,.
\tag{97}
$$

Eq. (97) is natural as the slow degrees of freedom are the longitudinal coordinates.

Note that these two equations admit a unique solution. Indeed, solving (96) for $A$ gives

$$
A = \frac{1}{2}C^{-2}\eta^{1-2\gamma}[H, S]
\tag{98}
$$

Plugging in (95), we find

$$
\tilde{\mathcal{L}}_H S = -C^{-1}\eta^{1-\gamma}M\,,
\tag{99}
$$

where $\tilde{\mathcal{L}}_H S \equiv \{H, S\} + \frac{1}{2}C^{-2}\eta^{1-2\gamma}[[S, H], H]$ is introduced in definition C.2. By Lemma C.3, (99) admits a unique solution.

Then

$$
\Phi^j_{11TT} = -C^{-1}\eta^{1-\gamma}\tilde{\mathcal{L}}^{-1}_H M^{(j)} - \frac{1}{2}C^{-3}\eta^{2-3\gamma}[H, \tilde{\mathcal{L}}^{-1}_H M^{(j)}]\,.
\tag{100}
$$

The other components are (see eqs. (67),(68),(74),(75),(83),(71))

$$
\Phi^{j,ik}_{02LL} = -(H^\dagger)^{jl}\partial^L_i H^{ln}P^{nk}_L
\tag{101}
$$

$$
\Phi^{j,ik}_{02LT} = -P^{jl}_L\partial^L_i H^{ln}(H^\dagger)^{nk}
\tag{102}
$$

$$
\Phi^{j,ik}_{11LL} = -C^{-1}\eta^{1-\gamma}(H^\dagger)^{jl}\partial^L_i H^{ln}P^{nk}_L
\tag{103}
$$

$$
\Phi^{j,ik}_{11TL} = -C^{-1}\eta^{1-\gamma}P^{jl}_L\partial^L_k H^{ln}(H^\dagger)^{ni}
\tag{104}
$$

$$
\Phi^{j,ik}_{02TT} = C\eta^{\gamma-1}\Phi^{j,ik}_{11TT} + C^{-1}\eta^{-\gamma}\Phi^{j,(ni)}_{11TT}H_{kn} = O\left(\eta^{\min\{0,1-2\gamma\}}\right)
\tag{105}
$$

$$
\Phi^{ij}_{20} = C^{-1}\eta^{1-\gamma}\Phi^{(ij)}_{11} + \frac{1}{2}C^{-1}\eta^{2-\gamma}\Phi^{ij}_{02} = C^{-1}\eta^{1-\gamma}\Phi^{(ij)}_{11}\,.
\tag{106}
$$

The only contributing term at leading order in $\eta$ to the limiting diffusion equation is (106). Splitting it into longitudinal and transverse components, we find:

$$\Phi_{20LL}^{j,ik} = C^{-1}\eta^{1-\gamma}\Phi_{11LL}^{j,(ik)} = -\frac{1}{2}C^{-2}\eta^{2-2\gamma}\left((H^\dagger)^{jl}\partial_i^L H^{ln}P_L^{nk} + (H^\dagger)^{jl}\partial_k^L H^{ln}P_L^{ni}\right) \quad (107)$$

$$\Phi_{20TL}^{j,ik} = C^{-1}\eta^{1-\gamma}\Phi_{11TL}^{j,(ik)} = -\frac{1}{2}C^{-2}\eta^{2-2\gamma}\left(P_L^{jl}\partial_k^L H^{ln}(H^\dagger)^{ni} + P_L^{jl}\partial_i^L H^{ln}(H^\dagger)^{nk}\right) \quad (108)$$

and, using (100) and (93) we have, in matrix notation,

$$\Phi_{20TT}^j = \frac{1}{2}C^{-1}\eta^{1-\gamma}(\Phi_{11TT}^j + (\Phi_{11TT}^j)^T) = -C^{-2}\eta^{2-2\gamma}\tilde{\mathcal{L}}_H^{-1}(P_T\partial_j^L H P_T)\,. \quad (109)$$

$\square$

To write things more compactly, the following Lemma will be useful:

**Lemma C.5.** *For any transverse symmetric matrix $T$:*

$$\Phi_{20TT}^j[T] = -\frac{1}{2}C^{-2}\eta^{2-2\gamma}M^{(j)}[\tilde{\mathcal{L}}_H^{-1}T]\,, \quad (110)$$

*Proof.* From (61), $\Phi_{20TT}$ is proportional to the symmetric part of $\Phi_{11TT}$ which was denoted by $S$ below Eq. (93) and satisfies Eq. (99). Therefore $\Phi_{20TT}$ also satisfies Eq. (99), up to an overall factor:

$$\tilde{\mathcal{L}}_H\Phi_{20TT}^j = -\frac{1}{2}C^{-2}\eta^{2-2\gamma}M^{(j)}\,. \quad (111)$$

Then, for any $T$:

$$(\tilde{\mathcal{L}}_H\Phi_{20TT}^j)[T] = -\frac{1}{2}C^{-2}\eta^{2-2\gamma}M^{(j)}[T]\,. \quad (112)$$

Moreover,

$$(\tilde{\mathcal{L}}_H\Phi_{20TT}^j)[T] = \Phi_{20TT}^j[\tilde{\mathcal{L}}_H T]\,. \quad (113)$$

Since the two above equations hold for any $T$ and $\tilde{\mathcal{L}}_H$ is invertible, this implies

$$\Phi_{20TT}^j[T] = -\frac{1}{2}C^{-2}\eta^{2-2\gamma}M^{(j)}[\tilde{\mathcal{L}}_H^{-1}T]\,, \quad (114)$$

which is the statement of the lemma. $\square$

To leading order in $\eta$ we then have, for a symmetric matrix $V$, $\partial^2\Phi[V] = \sum_{i,j=1}^D \tilde{\partial}_i\tilde{\partial}_j\Phi V_{ij}$, and thus

$$\partial^2\Phi[V] = -\frac{1}{2C^2}\eta^{2-2\gamma}(\nabla^2 L)^\dagger\partial^2(\nabla L)[V_{LL}]dt - \frac{1}{C^2}\eta^{2-2\gamma}P_L\partial^2(\nabla L)[(H^\dagger)V_{TL}]dt$$
$$- \frac{1}{2C^2}\eta^{2-2\gamma}P_L\partial^2(\nabla L)[\tilde{\mathcal{L}}_H^{-1}V_{TT}]\,, \quad (115)$$

where $V_{LL} = P_L V P_L$, $V_{TL} = P_T V P_L$, and $V_{TT} = P_T V P_T$ are transverse and longitudinal projections of $V$.

We also have, using (48), and to leading order in $\eta$,

$$\partial\Phi\tilde{\sigma}dZ = \Phi_{10}\sigma dZ + \eta\Phi_{01}\sigma dZ = (C^{-1}\eta^{1-\gamma} + \eta)P_L\sigma dW\,, \quad (116)$$

where $dW$ is a Wiener process. Applying Theorem B.4, and keeping into account that, to leading order in $dt$, $d[Z^i, Z^j] = \delta^{ij}dt$, we find

$$dY = (C^{-1}\eta^{1-\gamma} + \eta)P_L\sigma dZ - \frac{1}{2C^2}\eta^{2-2\gamma}(\nabla^2 L)^\dagger\partial^2(\nabla L)[\Sigma_{LL}]dt$$
$$- \frac{1}{C^2}\eta^{2-2\gamma}P_L\partial^2(\nabla L)[(H^\dagger)\Sigma_{TL}]dt - \frac{1}{2C^2}\eta^{2-2\gamma}P_L\partial^2(\nabla L)[\tilde{\mathcal{L}}_H^{-1}\Sigma_{TT}]dt\,, \quad (117)$$

where $\Sigma = \sigma\sigma^T$.

For $\gamma < \frac{1}{2}$, $\tilde{\mathcal{L}}_H$ reduces to the Lyapunov operator at leading order in $\eta$, i.e. $\tilde{\mathcal{L}}_H S = \{H, S\}$. For $\gamma > \frac{1}{2}$, from (55), it is easy to see that the role of the divergent term proportional to $\eta^{1-2\gamma}$, when acting $\tilde{\mathcal{L}}_H^{-1}$ on $S$, is to set to zero the off-diagonal entries of $S_{ij}$ at $O(\eta^{1-\gamma})$, i.e.

$$S_{ii} = -2C^{-1}\eta^{1-\gamma}\lambda_i M_{ii}, \qquad S_{i\neq j} = 0, . \tag{118}$$

Using Lemma B.5, we finally conclude the following Corollary, which is the formal version of Theorem 3.4 in the main text:

**Corollary C.6.** *Consider the stochastic process defined in Eq. (26) parametrized by $\epsilon_n$, with initial conditions $(\pi_0, w_0) \in U$, under assumptions 3.1 and 3.2. Fix a compact $K \subset U$. Then the conclusions of Theorem B.4 apply, and $Y(t)$ satisfies the limiting diffusion equation*

$$\begin{aligned} dY =&(\tfrac{1}{C}\eta^{1-\gamma} + \eta)P_L\sigma dW - \tfrac{1}{2C^2}\eta^{2-2\gamma}(\nabla^2 L)^\dagger \partial^2(\nabla L)[\Sigma_{LL}]dt \\ &- \tfrac{1}{C^2}\eta^{2-2\gamma}P_L\partial^2(\nabla L)[(\nabla^2 L)^\dagger \Sigma_{TL}]dt - \tfrac{1}{2C^2}\eta^{2-2\gamma}P_L\partial^2(\nabla L)[\tilde{\mathcal{L}}_{\nabla^2 L}^{-1}\Sigma_{TT}]dt \,, \end{aligned} \tag{119}$$

*where $W(t)$ is a Wiener process.*

Let us now see the special case of label noise. In this case $\Sigma = cH$, so that $\Sigma$ is only transverse. Moreover, using (55),

$$\tilde{\mathcal{L}}_H^{-1} H = \frac{1}{2} P_T \,, \tag{120}$$

and

$$dY = -\frac{1}{4C^2}\eta^{2-2\gamma}P_L\partial^2(\nabla L)[cP_T]dt = -\frac{1}{4C^2}\eta^{2-2\gamma}P_L\nabla\text{Tr}(c\partial^2 L)dt \,. \tag{121}$$

This proves Corollary 3.5 in the main text.

## D  EFFECTIVE DRIFT IN UV MODEL

We start with mean-square loss

$$L = \frac{1}{2P}\sum_{a=1}^P \|f(x^a) - y^a\|^2, \tag{122}$$

with the data covariance matrix $\Sigma$ as defined in the text. The trace of the Hessian on the zero loss manifold ($L(w^*) = 0$) is given explicitly by

$$\text{tr}H = \frac{1}{n}\left(m\text{Tr}\left(\Sigma V^\top V\right) + \text{Tr}\,\Sigma\,\text{Tr}\left(UU^\top\right)\right). \tag{123}$$

Taking gradients of this and plugging into Corollary 3.5, repeated in Eq. (121) leads to an explicit expression for the drift-diffusion along the manifold

$$dY = -\frac{\epsilon^2\eta^{2-2\gamma}}{4PC^2}\frac{2}{n}\hat{S}_L Y dt, \quad \hat{S} = \begin{pmatrix} \text{Tr}\Sigma\,\mathbb{1}_n & 0 \\ 0 & m\Sigma \end{pmatrix}, \quad \hat{S}_L = P_L\hat{S}P_L, \tag{124}$$

The simplification we cite in the main text in Sec. 3.4 is due to the fact that for input and output dimension $d = m = 1$, we have that $\Sigma = \text{Tr}\Sigma = \mu_2$, and $\hat{S}$ is proportional to the identity.

For matrix sensing, in order to compute the trace of the Hessian, we use (12) with $\xi_t = 0$ and with slightly different notation ($V^\top$ instead of $V$). The loss is then

$$L = \frac{1}{Pd}\sum_{i=1}^P \left(y_i - \text{Tr}(A_i UV^\top)\right)^2 \,, \tag{125}$$

and define the data covariance matrices

$$\hat{\Sigma}^1 = \frac{1}{P}\sum_{i=1}^{P} A_i A_i^\top \in \mathbb{R}^{d\times d}, \quad \hat{\Sigma}^2 = \frac{1}{P}\sum_{i=1}^{P} A_i^\top A_i \in \mathbb{R}^{d\times d}. \tag{126}$$

Then the trace of the Hessian is

$$\mathrm{Tr}H = \frac{2}{d}\mathrm{Tr}\left(\hat{\Sigma}^2 UU^\top + \hat{\Sigma}^1 VV^\top\right). \tag{127}$$

We find the noise function is

$$\sigma_{\mu i} = \frac{2}{Pd}\nabla_\mu f(A_i) \in \mathbb{R}^{d^4 \times P} \tag{128}$$

where $f(A) = \mathrm{Tr}(UV^\top A)$. Since the Hessian on the zero loss manifold is $H_{\mu\nu} = \frac{2}{Pd}\sum_i \nabla_\mu f(A_i)\nabla_\nu f(A_i)$, we see that $\sigma\sigma^T = (2/Pd)H$. Therefore, we get

$$dY = -\frac{\eta^{2-2\gamma}}{4C^2}\frac{4\epsilon^2}{Pd^2}\hat{S}_L Y dt, \quad \hat{S} = \begin{pmatrix} \hat{\Sigma}^2 & 0 \\ 0 & \hat{\Sigma}^1 \end{pmatrix}, \quad \hat{S}_L = P_L\hat{S}P_L \tag{129}$$

To get a crude estimate of the coefficient $C$ in the main text, we approximate the top eigenvalue of $\hat{S}$ with $\frac{1}{d}\mathrm{Tr}\hat{\Sigma}^1$. With this, we get for

$$\eta^{-2+2\gamma}\tau_2^{-1} = \frac{\epsilon^2}{C^2 Pd^2}\frac{1}{dP}\sum_{i=1}^{P}\mathrm{Tr}A_i A_i^\top \approx \frac{\epsilon^2}{C^2 Pd^2}d\langle a_{ij}^2\rangle = \frac{\epsilon^2}{C^2 dP}\langle a_{ij}^2\rangle. \tag{130}$$

Here we have denoted by $a_{ij}$ an arbitrary element of the data matrices $A$, with brackets signifying an average over the distribution of these elements. Assuming the fast initial fast remains the same, and $\tau_1^{-1} \approx (C/2)\eta^\gamma$, we get

$$C^3 = \frac{2\epsilon^2}{dP}\langle a_{ij}^2\rangle \tag{131}$$

For the values used in our experiments, this gives $C \approx 0.12 \times P^{-1/3}$.

# E  LINEARIZATION ANALYSIS OF MOMENTUM GRADIENT DESCENT IN THE SCALING LIMIT

Here we elaborate on the discussion in Sec. 3.3, providing derivations of various results. We take a straightforward linearization of the deterministic (noise-free) gradient descent with momentum. Working in the extended phase space $x = (\pi, w)$, the dynamical updates are of the form

$$x_{t+1} = x_t + F(x_t), \quad F(x_t) = \begin{pmatrix} (\beta - 1)\pi_t - \nabla L(w_t) \\ \eta(\beta\pi_t - \nabla L(w_t)) \end{pmatrix}. \tag{132}$$

The fixed point of the evolution $x^* = (0, w^*)$ will have the momentum variable $\pi = 0$, and the coordinate satisfying $\nabla L(w^*) = 0$. Linearizing the update (132) around this point

$$\delta x_{t+1} = J(x^*)\delta x_t, \quad J(x^*) = \begin{pmatrix} \beta & -\nabla^2 L(w^*) \\ \eta\beta & 1 - \eta\nabla^2 L(w^*) \end{pmatrix}. \tag{133}$$

Note $\nabla^2 L(w^*)$ is the Hessian of the loss function at the fixed point. The spectrum of the Jacobian can be written in terms of the eigenvalues of the Hessian $\lambda_i$. This is accomplished by using a straightforward ansatz for the (unnormalized) eigenvectors of the Jacobian $k^i = (\mu^i q^i, q^i)$, where $q^i$ are eigenvectors of the Hessian with eigenvalue $\lambda_i$. Solving the resulting coupled eigenvalue equations for eigenvalue $\kappa^i$:

$$1 - \eta\lambda_i + \eta\beta\mu^i = \kappa^i, \quad -\lambda_i + \mu^i\beta = \mu^i\kappa^i. \tag{134}$$

For a fixed $\lambda_i$, there will be two solutions given by

$$\kappa^i_\pm = \frac{1}{2}\left(1 + \beta - \eta\lambda_i \pm \sqrt{(1 + \beta - \eta\lambda_i)^2 - 4\beta}\right), \quad i = 1, ..., D \tag{135}$$

$$\mu^i_\pm = \frac{1}{2\beta\eta}\left(\beta - 1 + \eta\lambda_i \pm \sqrt{(1 + \beta - \eta\lambda_i)^2 - 4\beta}\right). \tag{136}$$

For the set of zero modes $\lambda_i = 0$, we get the following modes: $\kappa_+ = 1$, corresponding to motion only along $w$, with eigenvector $k^i = (0, q^i)$. In addition, there is a mixed longitudinal mode which includes a component of $\pi$ along the zero manifold $k^i = (\mu_- q^i, q^i)$, and has an eigenvalue $\kappa_- = \beta$.

On the zero loss manifold, we can assume the Hessian is positive semi-definite, and that the positive eigenvalues satisfy

$$0 < c_1 \leq \lambda_i \leq c_2. \tag{137}$$

for constants $c_1, c_2$ independent of $\eta, \beta$.

We now analyze the spectrum of the Jacobian one eigenvalue at a time, and then use these results to informally control the relaxation rate of off-manifold perturbations. It is useful first to consider the conditions for stability, i.e. $|\kappa^i| < 1$, which are stated in (138,139) below:

$$\text{Case 1}: \text{ If } \eta\lambda_i < (1 - \sqrt{\beta})^2, \text{ then } \kappa^i_\pm \in \mathbb{R} \text{ and } |\kappa^i_\pm| < 1 \text{ iff } 0 < \eta\lambda_i < 2(1 + \beta). \tag{138}$$

$$\text{Case 2}: \text{ If } \eta\lambda_i > (1 - \sqrt{\beta})^2, \text{ then } \kappa^i_\pm \in \mathbb{C} \text{ and } |\kappa^i_\pm| = \sqrt{\beta} < 1. \tag{139}$$

*Proof of Case 1:* The condition $\eta\lambda_i < (1 - \sqrt{\beta})^2$ implies $A^2 > 4\beta$, where $A = 1 + \beta - \eta\lambda_i$. The condition for stability then requires $-1 < \kappa_i < +1$. We satisfy both sides of this inequality:

If $1 + \beta - \eta\lambda_i = A > 0$, then $|\kappa^i_-| < 1$, and $\kappa^i_+ > 0$, so we simply require $\kappa^i_+ < 1$, i.e.

$$\kappa_i < +1 \tag{140}$$

$$\frac{1}{2}(A + \sqrt{A^2 - 4\beta}) < 1 \tag{141}$$

$$+ \sqrt{A^2 - 4\beta} < 2 - A \tag{142}$$

$$A^2 - 4\beta < 4 - 4A + A^2 \tag{143}$$

$$A = 1 + \beta - \eta\lambda_i < 1 + \beta \quad \Rightarrow \eta\lambda_i > 0 \tag{144}$$

If $1 + \beta - \eta\lambda_i < 0$, then $|\kappa^i_+| < 1$, and $\kappa^i_- < 1$, so we only require $\kappa^i_- > -1$

$$-1 < \kappa^i_- \tag{145}$$

$$-1 < \frac{1}{2}(-|A| - \sqrt{A^2 - 4\beta}) \tag{146}$$

$$|A| - 2 < -\sqrt{A^2 - 4\beta} \tag{147}$$

$$2 - |A| > \sqrt{A^2 - 4\beta} \tag{148}$$

$$A^2 - 4|A| + 4 > A^2 - 4\beta \tag{149}$$

$$|A| = -1 - \beta + \eta\lambda_i < 1 + \beta \quad \Rightarrow \eta\lambda_i < 2(1 + \beta) \tag{150}$$

*Proof of Case 2:* the condition $\eta\lambda_i > (1 - \sqrt{\beta})^2$ implies $A^2 < 4\beta$, where $A = (1 + \beta - \eta\lambda_i)$. The corresponding eigenvalues of the Jacobian can be written $\kappa_i^{\pm} = (1/2)(A \pm \sqrt{A^2 - 4\beta}) = (1/2)(A \pm i\sqrt{4\beta - A^2})$. Computing the absolute value then gives $|\kappa_i^{\pm}|^2 = (1/4)(A^2 + 4\beta - A^2) = \beta$ $\square$

These results show us when the GD+momentum is stable. Next, assuming stability, we want to estimate the rate of convergence to the fixed point. More precisely, we would like to determine the fastest mode as well as the slowest mode. To this end, we define two quantities

$$\rho_1 = \max\{|\kappa_{\pm}^i|\big|\ i = 1, ..., D,\ |\kappa_{\pm}^i| < 1\}. \tag{151}$$

$$\rho_2 = \min\{|\kappa_{\pm}^i|\big|\ i = 1, ..., D,\ |\kappa_{\pm}^i| < 1\}. \tag{152}$$

Using the explicit scaling for momentum, and in the limit of small learning rate, we prove the following for $\rho_1$:

**Lemma E.1.** *Let $\beta = 1 - C\eta^{\gamma}$, and $\eta$ sufficiently small:*

*For $\gamma < 1/2$, the condition for Case 1 (138) holds,*

$$\rho_1 \approx 1 - (c_1/C)\eta^{1-\gamma}, \tag{153}$$

$$\rho_2 \approx 1 - C\eta^{\gamma}. \tag{154}$$

.

*For $\gamma > 1/2$, the condition for Case 2 (139), and*

$$\rho_1 = \sqrt{\beta} \approx 1 - (C/2)\eta^{\gamma}, \tag{155}$$

$$\rho_2 = \beta = 1 - C\eta^{\gamma}. \tag{156}$$

.

*Proof:* For small $\eta$, we find that

$$\eta^{-1}(1 - \sqrt{\beta})^2 = \left(1 - \sqrt{1 - C\eta^{\gamma}}\right)^2 \approx C^2\eta^{2\gamma-1}/4. \tag{157}$$

When $\gamma > 1/2$, this expression tends to zero as $\eta \to 0$. Therefore, for sufficiently small $\eta$, the condition for Case 2 (139) will be satisfied and $|\kappa^i| = \sqrt{\beta}$ for all $\lambda_i > 0$. Since $\beta < \sqrt{\beta}$, this implies $\rho_1 = \sqrt{\beta}$ and $\rho_2 = \beta$. The scaling behavior in the Lemma follows by substitution.

For $\gamma < 1/2$, (157) diverges as $\eta \to 0$, which means Case 1 (138) will obtain for all $\lambda_i$. Next, since for small $\eta$, $1 + \beta - \eta\lambda_i > 1 + \beta - \eta c_2 = 2 - C\eta^{\gamma} - \eta c_2 > 0$, since $C$ and $c_2$ are order one constants, and $\eta \to 0$. In this case, the largest contribution from the nonzero eigenvalues $\lambda_i$ will come from $\kappa_+^i$. In particular, we find

$$|\kappa_+^i| \leq \frac{1}{2}\left(1 + \beta - \eta c_1 + \sqrt{(1 + \beta - \eta c_1)^2 - 4\beta}\right), \tag{158}$$

$$= \frac{1}{2}\left(2 - C\eta^{\gamma} - \eta c_1 + \sqrt{C^2\eta^{2\gamma} - 2(2 - C\eta^{\gamma})\eta c_1 + \eta^2 c_1^2}\right). \tag{159}$$

For $\gamma < 1/2$, we have the hierarchy $\eta^{\gamma} > \eta^{2\gamma} > \eta > \eta^{\gamma+1} > \eta^2$. This allows us to simplify the upper bound

$$\rho_1 = \max|\kappa_+^i| \leq 1 - \frac{c_1}{C}\eta^{1-\gamma} + O(\eta). \tag{160}$$

Next, we find a lower bound for $\rho_2$. This will be controlled by $\kappa_-^i$. We may use then that

$$\rho_2 = \min\{|\kappa_-^i|\} \geq \frac{1}{2}\left(1 + \beta - \eta c_2 - \sqrt{(1 + \beta - \eta c_1)^2 - 4\beta}\right), \tag{161}$$

$$= \frac{1}{2}\left(2 - C\eta^{\gamma} - \eta c_2 - \sqrt{C^2\eta^{2\gamma} - 2(2 - C\eta^{\gamma})\eta c_1 + \eta^2 c_1^2}\right) \approx 1 - C\eta^{\gamma} + O(\eta^{1-\gamma}) \tag{162}$$

Finally, note that since $\eta^{1-\gamma} < \eta^{\gamma}$, we have that $1 - \frac{c_1}{C}\eta^{1-\gamma} > 1 - C\eta^{\gamma} = \beta$, so indeed $\rho_2 < \rho_1$.

Equipped with the upper and lower bounds on the spectrum leads naturally to bounds on the relaxation rate. For the purely decaying modes at $\gamma < 1/2$, we use $\rho_2^t \leq |\delta x_t^T| \leq \rho_1^t$, $\delta x_t^T$ represents the projection of the fluctuations $\delta x_t$ onto the transverse and mixed longitudinal modes. After applying Eqs. (153) and (154), we arrive at the result quoted in the main text in Sec. 3.3. For $\gamma > 1/2$, the modes are oscillatory. However, the eigenvalues within the unit circle have norm either $\sqrt{\beta}$, $\beta$, as reflected in Eqs. (155) and (156). This implies that we can estimate the decay rate of the envelope of the transverse and mixed longitudinal modes in this regime, thereby arriving at second expression quoted in the main text in Sec. 3.3.

## F  LINEARIZED SGD AND ORNSTEIN-UHLENBECK PROCESS ON THE VALLEY

In this appendix, we provide a derivation of some of the statements quoted in Sec. 1.1. To get there, we start with the basic model for momentum SGD (1) but linearize around a point on the valley $w_0 \in \Gamma$ where $L(w_0) = \nabla L(w_0) = 0$. Let $w_k = w_0 + \delta w_k$, and define the Hessian $H(w_0) = \nabla^2 L(w_0)$. Then

$$\pi_{k+1} = \beta \pi_k - H(w_0)\delta w_k + \epsilon\sigma(w_0)\xi_k, \qquad \delta w_{k+1} = \delta w_k + \eta\pi_{k+1}, \tag{163}$$

Consider the projector along transverse nonzero eigenmode $\lambda$ of $H(w_0)$, $P_\lambda^T$, and define $P_\lambda^T x = X$, and $P_\lambda^T \pi = \Pi$. Let $\bar{\sigma} = P_\lambda^T \sigma$, and $P_\lambda^T H = \lambda P_\lambda^T$. Let $\bar{\sigma}\bar{\sigma}^\top = \Lambda$. Then

$$\Pi_{k+1} = \beta\Pi_k - \lambda X_k + \epsilon\bar{\sigma}(w_0)\xi_k, \tag{164}$$

$$X_{k+1} = X_k + \eta\Pi_{k+1}, \tag{165}$$

$$= X_k + \eta\beta\Pi_k - \eta\lambda X_k + \eta\epsilon\bar{\sigma}(w_0)\xi_k \tag{166}$$

This is a simple OU process, and we can easily compute the second moments. Define the second moments

$$C_{12}(k) = \langle X_k\Pi_k\rangle, \quad C_{11}(k) = \langle X_kX_k\rangle, \quad C_{22}(k) = \langle\Pi_k\Pi_k\rangle. \tag{167}$$

We find by taking the equations above, squaring them, then averaging over the noise,

$$C_{22}(k+1) = \beta^2 C_{22}(k) - 2\beta\lambda C_{12}(k) + \lambda^2 C_{11}(k) + \epsilon^2\Lambda, \tag{168}$$

$$C_{12}(k+1) = \eta\beta^2 C_{22}(k) + \beta(1 - 2\eta\lambda)C_{12}(k) - \lambda(1 - \eta\lambda)C_{11}(k) + \eta\epsilon^2\Lambda, \tag{169}$$

$$C_{11}(k+1) = \eta^2\beta^2 C_{22} + 2\eta\beta(1 - \eta\lambda)C_{12}(k) + .(1 - \eta\lambda)^2 C_{11}(k) + \eta^2\epsilon^2\Lambda. \tag{170}$$

Next, assuming a stationary distribution implies $C(k+1) = C(k)$, which then allows us to solve for equilibrium variance, and extract the main quantity of interest, which is the variance of the weights. We find then

$$C_{11}(k) = \frac{\eta^2\epsilon^2\Lambda}{(1-\beta)\eta\lambda(2(1+\beta) - \eta\lambda)}. \tag{171}$$

In the limit of small $\eta$ we extract the scaling behavior quoted in Sec.(1.1).

We can see how the mixing timescale $\tau_1$, discussed in Sec. 1.1, arises from this linearized analysis. By taking the expectation value of the OU process, the noise will vanish and we find that the average values follow the linearized GD dynamics analyzed in E. Thus, from this linearized GD analysis we can extract the characteristic timescale for the OU process to approach its mean value.

