# OpenReview forum: "Flatter, Faster: Scaling Momentum for Optimal Speedup of SGD"
_ICLR.cc/2023/Conference — Submitted to ICLR 2023_

### Official Review · Reviewer_e9QR · 2022-10-19

**Confidence:** 4
**Correctness:** 2
**Technical Novelty And Significance:** 2
**Empirical Novelty And Significance:** Not applicable
**Recommendation:** 3

**Clarity, Quality, Novelty And Reproducibility:**

[clarity] the clarity of the paper should be improved
[quality] several key claims are not well justified

**Strength And Weaknesses:**

First, I have to mention that the paper is not well written, and it takes a lot of effort for readers to understand. The authors should consider improving the clarity of the paper. I list a few points below:

> There should be definition or clarification of notations before using them. For example, in the 5th line of Sec 3.1, “the parameter $\epsilon > 0$ controls the size of the noise …”. However, there is no $\epsilon$ mentioned before. Hence, it is hard to understand what this line is talking about. In addition, in Assumption 3.1, $C^3$ appears before the notation definition of $C^n$. Furthermore, in Sec 3.3, without defining $\delta x$, it is hard to understand the equation $\delta x_{t+1} = J(x^*)\delta x_t$.

> The paper does not set up the problem well. The paper has some settings in the introduction, Sec 1.1, and some other in Sec 3.1. None of the subsections established the problem setting; instead, the two sections are referring to each other, and readers have to go back and forth to get the context.

> There should be a discussion of the connection between the time scaling analysis and the ordinary SGD setting. In machine learning, SGD is an iterative algorithm taking on discrete time steps with predefined learning rates. It is important to clarify what the timescale means to SGD, and why there is a parameter $\epsilon$ controlling the size of noise and can be taken to zero.

Second, the modeling of the momentum SGD using Eq.(1) is not appropriate. As in practice, the noise of SGD arises from the estimation of gradients via samples, instead of an additive noise as in Eq.(1). The difference in how the noise is generated results in different properties of the noisy gradient. For example, in the over-parameterized case, the stochastic gradient has been found to have a vanishing noise as the algorithm converges to a global minimum, see automatic variance reduction in [1]. However, the additive noise often keeps a certain level of noise even if it is close to convergence. In addition, it has been found that the variance of stochastic gradients cannot be bounded by a constant (see [2]), while the additive noise usually can.
Hence, it is better to start the analysis from following equation (instead of Eq.(1)): $\pi_{k+1} = \beta \pi_k - \nabla l_i(w_k)$, where $l_i$ is the loss function evaluated on one training sample.

Third, I could not understand why the $\approx$ holds in Eq.(2). In principle, the momentum $\pi_{k+1}$ should contain gradient noise, and should not be averaged over. Removing the noise is critical to the final conclusion. I hope the authors can explain this.

Fourth, the paper does not explain why Eq.(5) solves Eq.(1). This is a key step, and should not be omitted from the paper.

Fifth, the analysis is conducted assuming in the vicinity of the global minima, namely very close to the solution. Many of the assumptions do not hold outside of the vicinity, and the analysis does not apply (for example, the dominance of perpendicular displacement over the longitudinal). Hence, it is possible that the theoretical optimal power of $\gamma=2/3$ is not optimal outside of the vicinity.

Overall, as for the $\gamma=2/3$ rule, the theoretical analysis in Sec 3.3 is not rigorous (as admitted informal by the authors), but it experimentally works well (as in the ResNet). In addition, as mentioned above in the fifth point, the theory applies to the vicinity of the global minima, and does not match the experimental setup. Given these, I would consider the $\gamma=2/3$ rule as a good finding that is worth trying by practitioners, but I don’t think the theory supports it well.


[1] Accelerating SGD with momentum for over-parameterized learning

[2] Better Theory for SGD in the Nonconvex World


**Summary Of The Paper:**

This paper studies the convergence of momentum SGD at the vicinity of global minima in the over-parameterized setting. With some approximations which only hold in this vicinity, the scaling analysis predicts an exponent $\gamma=2/3$ as the scaling of the momentum parameter that maximally accelerates the training.

**Summary Of The Review:**

The the paper is not written clearly, and it took much effort for reading.

Several key claims are not well supported, with the proof or justification missing. Without them, I am susbicious of the correctness of the results.

The analysis is conducted within the vicinity of global minima, outside of which many assumptions or treatments do not hold. However, in practice most of the optimization happens outside of this vicinity. Hence, I doubt whether the results of the paper is applicable in practice.

---

> ### Author Response · Authors · 2022-11-19
> **Response to Reviewer e9QR (Part 1 of 2)**
>
> We thank the referee for their feedback, especially regarding their comments on the presentation of our framework. We made several improvements in the explanation of the problem setup which we believe should address the referee's questions. Below is a detailed reply to the referee's points and a description of our edits.
>
> 1. **Improved presentation:** We introduced the parameter that controls the strength of the noise, $\epsilon$, in eq. (1). We also explicitly incorporated $\epsilon$ in the heuristic scaling analysis of Sec. 1.1. We moved the definition of notations to the paragraph above Sec. 1.1 and used boldface to facilitate its location. In Sec. 3.3, we added the explicit definition of $\delta x=x_t-x_0$.
>
>     We streamlined the presentation of the problem setup. We rewrote Sec. 1.1 putting more emphasis on the assumptions and the key logical implications, breaking some of the algebra into smaller steps and explaining them in words. We note that Sec. 1.1 provides a heuristic argument for our proposed scaling, as the title suggests it should be not regarded as a rigorous derivation. Separately, we also added Sec. 1.2, which is new and summarizes the basic building blocks of the framework we use in Secs. 3.1,3.2 in order to derive the limit drift-diffusion dynamics along the manifold formed by the global minima. In Sec. 1.2 we reformulate the main assumptions in the language of the framework in Sec. 3. The main point of Sec. 1.2 is that it introduces a projector $\Phi$ that systematically isolates the slow-moving degrees of freedom. We added this section inspired by the referee's comment, and believe it should help clarifying the setting of Sec. 3.
>
>     We included a discussion in Sec. 1.1 that explains how the scaling analysis is related to SGD time steps. In our paper, we consider two timescales which are related to the longitudinal and transverse dynamics happening near the manifold of global minimizers $\Gamma$. In the transverse direction, the weights satisfy an Ornstein-Uhlenbeck process, and we call $\tau_1$ the mixing time of this process, which is the number of SGD time steps it takes for the displacements transverse to $\Gamma$ to approach equilibrium.
>
>     The other important timescale we consider is $\tau_{2}$, which describes the motion in the direction tangential to $\Gamma$. $\tau_2$ equals the number of SGD time steps it takes so that the displacement of $w_k$ along $\Gamma$ becomes a finite number that does not vanish when we take $\epsilon\to0$ first, and $\eta\to 0$ afterward.
>
>
>     Finally, the new Sec. 1.2 explains the role of the parameter $\epsilon$ that controls the size of the noise. The limit $\epsilon \to 0$ is necessary in order to find explicitly the limit drift-diffusion equation. The conditions under which this limit can be taken were originally formulated in [1], and are reviewed in Appendix B. In practice, this limit signifies that the prediction of Theorem 3.4 becomes increasingly better as the strength of label noise is made smaller and the minibatch size is made larger. We included this comment in the last paragraph of p. 2.
>
>
> 2. **Modeling SGD with eq. (1):** We respectfully disagree with the referee on this point. The noise function $\sigma(w)$ in Eq. (1) does depend on the weights $w$, and this dependence models the gradient noise due to SGD with label noise. This is an established approach to study SGD that has been extensively used in the literature (see e.g. [2]). We thank the referee for pointing two relevant references, which we included below eq. (1). Additionally, we would like to remark that one important advantage of label noise, where we believe our scaling prediction is mostly relevant (as per Corollary (3.5)), is to compensate for automatic variance reduction in that, even when the data is overfitted, label noise will continue to drive the model towards wider minima of the loss landscape, as appreciated e.g. in [3].
>
> 3. **Gradient noise in eq. (2):** We thank the referee for pointing this out, which helped improving our presentation. We replaced the former Eq. (2) with the inline equation "$P_L(\langle \delta w_{k+1}\rangle-\langle\delta w_k\rangle)=P_L\langle\pi_{k+1}\rangle$'' which is immediately implied by Eq. (1), with no approximation. The step ``$\approx$'' in Eq. (2)  of the previous version of the manuscript implicitly took into account that fluctuations of $\pi_{k+1}$ around its mean, projected in the longitudinal direction, do not matter for the argument: this is based on the intuition that the weights undergo a biased average drift along the manifold $\Gamma$. We agree that this step was not sufficiently clear, and believe that the above substitution will provide a more transparent step for the discussion.

---

> > ### Comment · Reviewer_e9QR · 2022-12-08
> > **after rebuttal**
> >
> > I thank the authors for the detailed explanation and improving the paper. I acknowledge the authors' efforts in improving the presentation of the papers. The overall presentation is much better in the revision.
> >
> > I still have the following concerns:
> >
> > > As the authors mentioned, the analysis relies on taking $\epsilon \to 0$ first, and $\eta \to 0$ afterward. However, in SGD the noise level $\epsilon$ is controlled by the learning rate $\eta$ (for example, see the automatic variance reduction); that is to say, you can not take $\epsilon \to 0$ before $\eta \to 0$. Hence, I have doubts in the applicability of this analysis onto (momentum) SGD.
> >
> > > The paper also assumes that the algorithm should be initialized in the vicinity of the solution manifold. Especially, $\pi_0 = 0$ near the solution manifold. But, in practice, when SGD is close to the manifold, the momentum is usually not zero, but an (decayed) accumulation of gradient history. Hence, I could not agree with the applicability of this analysis onto the SGD "after the training loss approaches zero".
> >
> > > I am also not clear how large the $\epsilon$ and training loss could be so that the analysis is valid, as the analysis is often taking to the limit. In fact, in most machine learning tasks, we finish training before an exact zero training loss (e.g., finish once training loss is less than a small number, say $10^{-6}$). It is not clear to me that whether the allowed $\epsilon$ or training loss in the analysis of the paper is too small such that there is no overlap with the practical training. In case not, the analysis, especially the $2/3$ power law, would be not useful for the practical SGD.
> >
> > > It is better to be more specific on (and give some references for) "Deep neural networks typically posses a manifold of parametrizations with zero training error". I believe this is almost true for over-parameterized setting, but not necessarily for under-parameterized setting. See for example, Liu, et al, "Loss landscapes and optimization in over-parameterized non-linear systems and neural networks". It says that under-parameterized models can still have isolated local/global minima.

---

> > > ### Author Response · Authors · 2022-12-09
> > > **Re: after rebuttal**
> > >
> > > We are happy that the reviewer is satisfied with the presentation of the revised manuscript. Below we address point by point the comments raised in the last post by the reviewer:
> > >
> > > 1. We agree that in the case of SGD, the learning rate controls the noise level. However, as we explain in the revised manuscript, our analysis relies on taking the limit $\epsilon\to 0$ by increasing the minibatch size and reducing the label noise, while keeping the learning rate fixed. In eq. (1), the learning rate $\eta$ appears only in the second equation, which describes the update of the model parameters. The last term of the first equation, $\epsilon\sigma(w_k)\xi_k$, comes from sampling the gradient of the loss, and is thus independent of the learning rate $\eta$. This allows us to decouple the noise level from the learning rate, making it possible to take the limit $\epsilon\to 0$ first and then take the limit $\eta\to 0$. As a result, our analysis is still applicable to SGD.
> > >
> > > 2. We agree that in practice, the momentum variable may be initially quite large when the weights approach the zero-loss manifold. However, as the gradients become smaller near the manifold, the momentum variable will also decay and approach a value controlled by the asymptotic size of the gradients, which will be small. We would like to emphasize that Theorem 3.2 is valid for any initial condition as long as SGD converges to the valley, which is a rather general setting.
> > >
> > > 3. We thank the reviewer for raising this question. In our experiments on CIFAR10, we used a relatively large label noise value of $p=0.2$, and the training loss was roughly $10^{-1}-10^{-3}$ throughout the experiment. These values allowed us to obtain consistent results with the $2/3$ power law prediction, as we showed in the plots of Fig. 1 and Fig. 2. This indicates that, in practice, there is a wide regime of validity for our predictions.
> > >
> > >      Heuristically, the effect of large $\epsilon$ and nonzero loss can be appreciated by inspecting the derivation outlined in Sec. 1.1. As illustrated there, the longitudinal drift is induced by the third derivative of the loss. Therefore, if the gradient of the training loss is much smaller than the third derivative contribution, the dynamics will still be dominated by the noise, and our analysis should still hold. Additionally, we also require that the noise $\epsilon$ is not too large, as this would cause large transverse displacements from the valley, thus activating the fourth derivative term in the Taylor expansion of the loss when estimating the longitudinal displacements. This correction ultimately leads to  deviations from the scaling behavior we found.
> > >
> > > 4. We thank the reviewer for bringing up this interesting reference. We will cite it, and clarify in the sentence quoted by the reviewer that we are referring to overparameterized neural networks.

---

> > > > ### Comment · Reviewer_e9QR · 2022-12-11
> > > > **Reply**
> > > >
> > > > I could not quite agree with some of the explanations above.
> > > >
> > > > 1. In this analysis, I think it is invalid to take the limit of large minibatch size in SGD to control the stochastic noise $\epsilon$.  SGD usually has a fixed batch-size, and the level of stochastic noise $\epsilon$ is only determined by how close it is from the solution. Note that different batchsizes of SGD essentially correspond to different algorithms.
> > > >
> > > > 2. I think the authors have to be careful when talking about "small", because that depends on what do you compare with. We only know the momentum could be smaller near the manifold than far away from the manifold. But, it does not neccessarily mean whether it is small enough for this vicinity analysis. Theoretically speaking, the theory assumes momentum be exactly zero, so as long as momentum is non-zero (regardless of how small it is), the analysis does not apply.
> > > >
> > > > 3. Although the authors give some experiments for illustration, I still want to see **in theory** how large the $\epsilon$ and training loss could be so that the analysis is valid.

---

> > > > > ### Author Response · Authors · 2022-12-12
> > > > > **Reply (Part 1 of 2)**
> > > > >
> > > > > We thank the reviewer for their feedback. Below is our reply.
> > > > >
> > > > > 1. We agree that the noise level $\epsilon$ is typically fixed in SGD and that different batch sizes correspond to different algorithms.
> > > > > Theorem 3.4 concerns a sequence of stochastic processes, each one at a \emph{fixed} value of $\epsilon$, so each element of the sequence should be regarded as a different algorithm. The theorem states that, in taking the limit of this sequence, the latter converges to the solution of the drift-diffusion equation. Therefore, our analysis is valid when the batch size is not allowed to vary.
> > > > >
> > > > >
> > > > > 2. We appreciate the reviewer's concern about having a theoretical comparison for ``small'' size. With regard to the reviewer's comment about the assumption of zero momentum in the theory, we would like to clarify that our analysis does not require the momentum to be exactly zero. Indeed if it were zero the SGD update equations imply that the evolution would stop. The main Theorem 3.4 holds for a generic initial condition, which includes the case where momentum is nonzero on the valley. The heuristic analysis in section 1.1 illustrates this point by showing that the variance of the transverse component of momentum is $\Theta(\epsilon^2)$ and the average of the longitudinal component is $\Theta(\epsilon^2)$. This demonstrates that our analysis does not require the momentum to be exactly vanishing and, in fact, it implies it to be nonzero.

---

> > > > > ### Author Response · Authors · 2022-12-12
> > > > > **Reply (Part 2 of 2)**
> > > > >
> > > > > 3. Determining rigorous bounds on the noise $\epsilon$ and the loss function goes beyond the mathematical framework of [1], on which the theoretical part of this paper relies, and is an important open problem. Additionally, to write these bounds in terms of model hyperparameters requires more information about the specific model architecture. It would be interesting to further develop this direction in future work and carefully derive rigorous bounds on the size of $\epsilon$ and the training loss, which would be beyond the scope of this paper.
> > > > >
> > > > > Below we include an extended version of our previous reply to point 3, where we illustrated how increasing $\epsilon$ and loss function may lead, eventually, to the breakdown of our analysis. We emphasize that the discussion below is very qualitative and is not intended to be a rigorous derivation.
> > > > >
> > > > > Following the Taylor expansion given at the end of p. 2 of the manuscript, now without restricting the expansion point $w_0$ to lie strictly on a valley, the dynamics of the displacements $\delta w_k$ are governed by
> > > > >
> > > > > ${\displaystyle (1)  \qquad \nabla L(w_0)+\nabla^2 L(w_0)[\delta w_k]+\frac 12\partial^2(\nabla L(w_0))[\delta w_k,\delta w_k]}$
> > > > >
> > > > > We assume that the Hessian $\nabla^2 L(w_0)$ is still semipositive definite and such that its eigenvalues can be split into two subsets $\{\lambda^T_i:i=1,\dots, D-M\}$ and $\{\lambda^L_j:j=1,\dots M\}$ (see Assumption 3.1 for definition of $D$ and $M$) such that $\lambda^L_j/\lambda^T_i\ll 1$ for any $i$ and $j$. This way we can still define approximate longitudinal and transverse displacements $\delta w_k^L$ and $\delta w_k^T$, respectively. We now obtain a heuristic upper bound on the gradient $\nabla L(w_0)$  for which we expect intuitively that our analysis applies (note there is no general upper bound on the loss $L$ itself, e.g. we can shift it by a constant $L\to L+c$ and the SGD update will remain the same). Let us first project expression (1) above onto the transverse direction. Recall from Sec. 1.1 that $\delta w_k^T=\Theta(\epsilon)$, so, for this relationship to still hold at finite $\nabla L(w_0)$, we need the transverse part of $\nabla L(w_0)$ to be smaller than the second term in expression (1) (as the latter controls the dynamics of $\delta w_k^T$, as was discussed in Sec. 1.1) which means, using the notation of the paper, $(\text{Id}-P_L)\nabla L(w_0)=o(\epsilon)$. This can be regarded as a heuristic upper bound on the transverse gradient of the loss. Next, projecting expression (1) onto the longitudinal direction, the first term should be smaller than the third term, as the first term produces longitudinal deterministic motion that is not induced by the noise, unlike the third term (thus breaking the condition that longitudinal displacements are driven by the noise). This implies that the first term should be not bigger than $P_L\nabla L(w_0)=o(\epsilon^2)$, which can be regarded as an upper bound on the longitudinal gradient. A similar reasoning can be applied to the second term of expression (1).
> > > > >
> > > > > Separately, we now attempt to estimate an upper bound on the noise $\epsilon$. If $\epsilon$ is large, then the transverse displacements $\delta w_k^T$ will also be large, and the Taylor expansion in expression (1) should include higher order terms, e.g. the fourth derivative of $L$. Such higher-order term comes with a different power of $\epsilon$ (because it is proportional to $\delta w_k^3$), and we should make sure that it does not grow larger than the third-derivative term, otherwise this will affect the way the longitudinal drift scales with $\epsilon$. Schematically, taking $\delta w_k\sim \epsilon$, this means that we want $\partial^3 L\cdot \epsilon^2>\partial^4 L\cdot \epsilon^3$. Since we do not have control of the fourth derivatives of the loss $L$, this condition should be interpreted as an upper bound on $\epsilon$, suggesting a bound of the form $\epsilon<(\partial^3L) / (\partial^4 L)$. If this condition is not satisfied, the higher powers in $\epsilon$ contributing to expression (1) will become important, and this will spoil our scaling analysis as well as Theorem 4.3. In particular, the statement about the longitudinal timescale $\tau_2=\Theta(\epsilon^2)$ will cease to be valid.
> > > > >
> > > > > [1] Katzenberger, Gary Shon, Solutions of a stochastic differential equation forced onto a manifold by a large drift. The University of Wisconsin-Madison, 1990.

---

> ### Author Response · Authors · 2022-11-19
> **Response to Reviewer e9QR (Part 2 of 2)**
>
>
> 4. **Equivalence of (old) eq. (5) and eq. (1):** We appreciate the question from the referee and include a full proof in appendix B.2. This proof follows from the definitions in [1]. In short, the old eq. (5) (now eq. (4)) is the integral form of the SGD equations. Viewing SGD as a ``differential equation with delta function updates," in a way made rigorous in [1], we can see eq. (5) as just integrating both sides of the update equation. Additionally, in response to this question we have added section 1.2 which we hope will make the transition between the discrete formulation and the continuous one smoother for the reader.
>
>
> 6. **Analysis in the vicinity of global minima:** We agree with the referee that our proposed scaling rule has theoretical control only in the vicinity of the valley. In fact, we would like to remark that significant improvement of test error can happen after the training loss approaches zero. For example, [3] proves that in matrix sensing, SGD with label noise yields the ground truth rather than any other interpolating solution, and this happens after SGD approaches a particular interpolating solution that is not the ground truth. They obtain similar results for 2-layer ReLU networks.
>
>     Additional examples of improvement in test accuracy after the weights reach a region of small loss include epoch-wise double descent as in Fig. 10 and 18 of [4], and learning of algorithmic datasets as in Fig. 1 of [5]. In light of this specific setting, we have included a line in our introduction specifying that we are interested only in training after the training loss becomes small to help with the clarity of our exposition.
>
>
> 7. **Practical implications of the $\gamma = \frac{2}{3}$ rule:** We are glad that the referee appreciates the relevance of our prediction for future empirical studies. We are confused by the referee's comment about the inapplicability of our theoretical framework to the experimental setup. We understand the referee's comment as saying that the assumptions of the theory are different from the experimental setup. The main assumption in our theory is that the weights move along a manifold of global minima. In all our experiments, most of training happens at a very small training loss, while test loss has finite value and continues smoothly to decrease. In Appendix A.2 of the current version of the manuscript, Fig. 3b), we included a plot of the trace of the Hessian showing that it decreases along training (a similar plot for Resnet18 on CIFAR10 was obtained in [6]). We take these facts as very compelling evidence that the weights are moving close to a zero-loss manifold.
>
> References:
>
> [1] Katzenberger, Gary Shon, Solutions of a stochastic differential equation forced onto a manifold by a large drift. The University of Wisconsin-Madison, 1990.
>
> [2] Li, Qianxiao, et al., Stochastic modified equations and adaptive stochastic gradient algorithms. In International Conference on Machine Learning (pp. 2101-2110). PMLR, 2017.
>
> [3] Blanc, Guy, et al., Implicit regularization for deep neural networks driven by an ornstein-uhlenbeck like process. Conference on learning theory. PMLR, 2020.
>
> [4] P. Nakkiran, et al, Deep
> double descent: Where bigger models and more data hurt. ICLR, 2020.
>
> [5] Power, Alethea, et al., Grokking: Generalization beyond overfitting on small algorithmic datasets. arXiv preprint arXiv:2201.02177 (2022).
>
> [6] Damian, Alex, et al., Label noise sgd provably prefers flat global minimizers. Advances in Neural Information Processing Systems 34 (2021): 27449-27461.

---

> > ### Author Response · Authors · 2022-12-05
> > **Happy to provide additional clarification**
> >
> > We hope our response clarified your initial concerns and questions. We would be happy to provide further clarifications where necessary.

---

### Official Review · Reviewer_F44e · 2022-10-26

**Confidence:** 3
**Correctness:** 4
**Technical Novelty And Significance:** 3
**Empirical Novelty And Significance:** 1
**Recommendation:** 6

**Clarity, Quality, Novelty And Reproducibility:**

Analysis of SGDM with this SDE while their proposed scaling regime is novel. All the claim in the paper has theoretical justification.

**Strength And Weaknesses:**

Strength:
 This paper analyzes the SDE for SGD with momentum and label noise and tries to show that adding momentum accelerates the convergence but not hurting the generalization. This is an important challenge while training big models with numerous training datasets such as DNN. Their analysis provides a theoretical basis to set the momentum hyperparameter.

Weakness:
My major concern with regard to the paper is its writing. Mainly the concepts and findings are not explained well formally. It could be more readable if they explain more about the notation they use and also not overloading.


**Summary Of The Paper:**

This paper analyzes the SDE for SGD with momentum and label noise and tries to show that adding momentum accelerates the convergence but not hurting the generalization. This is an important challenge while training big models with numerous training datasets such as DNN. They show that there is an interplay between the speedup of momentum and the limiting diffusion generated by SGD noise. Using that they show two characteristic timescales associated with the training dynamics, and the longest timescale essentially governs the training time. Maximum acceleration is thus reached when these two timescales coincide, which leads to identifying an optimal scaling of the hyperparameters.

**Summary Of The Review:**

1- My major concern with regard to the paper is its writing. Mainly the concepts and findings are not explained well formally. It could be more readable if they explain more about the notation they use and also not overloading.

2- They show that for specific \gamma= ⅔ it SGD with momentum (SGDM) can reach best performance. However, their empirical results don’t provide strong evidence. It would be helpful to run SGDM with the proposed hyperparameter over several DNN models and compare the test accuracy with SGD and other popular optimization methods.

3- For the analysis of the phase when SGDM moves toward zero loss valley they assume GDM instead of SGDM. However, there is no enough justification for why this is true.

4- In eq. 2 no justification for why the last approximation is true.

5- In eq. 3, since we are in zero valley, isn’t \nalba L (w_i) = 0? But the approx in the next line is not zero.

---

> ### Author Response · Authors · 2022-11-19
> **Response to Reviewer F44e**
>
> We thank the referee for the positive feedback on our manuscript, and for their feedback that helped improving presentation and content of the paper. We believe we addressed the referee's requests, as detailed in the points below:
>
> 1. **Improved presentation:** We appreciate the referee's feedback on the quality of the presentation of concepts and notation. We moved the notation to the last paragraph before Sec. 1.1, and emphasized in boldtext the word "Notation,'' which will help in guiding the reader. Additionally, we substantially rewrote Sec. 1.1 clarifying the setup and the heuristic steps that lead to our proposed scaling rule. In particular, what used to be Eqs. (2) and (3) have been broken in smaller steps, emphasizing the assumptions and significance of each step, and slightly simplifying the logic. We also added a new subsection, Sec. 1.2, where we included an informal discussion of the general framework used in Secs. 3.1 and 3.2 which is a key part of the theoretical analysis. We also used Sec. 1.2 to restate our assumptions within the language of this framework.
>
> 2. **Empirical work:** We appreciate the referee's keen interest in the practical implications of our algorithm. To provide further evidence for the applicability of our theory we performed experiments with a 6-layer MLP on FashionMNIST, both a new architecture and dataset. The results of this experiment can be seen in appendix A.4. Again we find that the experimentally extracted scaling is consistent with our theory.
>
>     In Appendix A.2, we included a comparison of performance in test loss for different values of momentum hyperparameter $\beta$. This clearly shows a non-monotonicity of the performance in $\beta$, i.e. there exists an intermediate value of $\beta$ that maximizes speed-up. This plot shows the raw data we use to find the optimal $\beta$ value as a function of learning rate. More broad comparisons to the state-of-the-art optimization algorithms will have to come in the form of future work, as in the present paper we are concerned with the phase of training which follows train loss convergence. Therefore, we believe training a model start-to-finish with an optimal momentum hyperparameter is outside the scope of this paper and will be an exciting direction which we plan to investigate in follow-up work.
>
> 3. **GDM in place of SGDM:** We believe the referee is referring to the analysis of Sec. 3.3. We would like to clarify that, to study convergence to the valley, SGDM can be approximated by GDM only in the limit $\epsilon\to 0$. The purpose of this section is to estimate the timescale associated to the relaxation of the weight displacements that are transverse to the valley, which we called $\tau_1$, as $\epsilon\to 0$. Since the Hessian governing the transverse dynamics is nondegenerate, we expect $\tau_1$ to be finite as $\epsilon\to 0$. In the paper we are interested in the leading order expression of $\tau_1$ in this limit, therefore it is sufficient to compute $\tau_1$ by considering the noiseless dynamics $\epsilon=0$. We included a paragraph clarifying this point in the beginning of Sec. 3.3.
>
> 4. **Approximation in eq. (2):** We thank the referee for bringing this to our attention, and helping improve our presentation. We have replaced the former Eq. (2) with an inline equation $P_L (\langle \delta w_{k+1}\rangle - \langle \delta w_k \rangle) = P_L\langle \pi_{k+1}\rangle$ which is immediately implied by Eq. (1) without approximation. The step ``$\approx$'' in Eq. (2)  of the previous version of the manuscript implicitly took into account that fluctuations of $\pi_{k+1}$, projected in the longitudinal direction, do not matter for the argument: this is based on the intuition that the weights undergo a biased average drift along the manifold $\Gamma$. We agree that this step was not sufficiently clarified, and replaced the approximation step in Eq. (2) with the equality mentioned above, where we took the average on both sides.
>
> 5. **Being at the valley in equation 3:** While we are close to the valley, and therefore the perpendicular displacement has zero mean, the latter still has finite variance. In the former eq. (3) the term proportional to the mean $\langle v^T_k \rangle$ vanishes whereas it's variance remains finite due to the effect of noise. Along with the stationarity approximation, this allows us to perform the final step with ``$\approx$.''

---

> > ### Author Response · Authors · 2022-12-05
> > **Happy to provide additional clarification**
> >
> > We hope our response clarified your initial concerns and questions. We would be happy to provide further clarifications where necessary.

---

### Official Review · Reviewer_oJpk · 2022-11-02

**Confidence:** 3
**Clarity, Quality, Novelty And Reproducibility:** The quality of presentation seems to …
**Correctness:** 4
**Technical Novelty And Significance:** 3
**Empirical Novelty And Significance:** 2
**Recommendation:** 6

**Strength And Weaknesses:**

Strength:
- Understanding the best momentum setting is critical in neural network optimization
- The discovery of the optimal momentum scaling, and its connection to finding flatter minima are very interesting results
- Solid theoretic results with clear motivations

Weakness:
- In addition to identifying the best momentum setting, could the authors also provide some theoretic insight of the momentum acceleration? Such as discussing the effect of $\beta > 0$ compared with $\beta = 0$.
- This work would benefit from more empirical evaluations to demonstrate the practical implication of the discovered scaling rule
- Is the label noise of SGD only an analytic artifact? Is there any empirical reason to add such label noise?
- The scaling rule $\beta = 1 - C\eta^\gamma$ seems arbitrary. Is there any reason for such a choice?

**Summary Of The Paper:**

This work extends the analysis framework of (Blanc et al., 2020) and (Li et al., 2022) to study the dynamic of heavy ball momentum along a manifold of minimizers. The authors proposed a momentum scaling rule $\beta = 1 - C\eta^\gamma$ and discovered the optimal scaling power as $\gamma=2/3$ under the analysis framework. The authors also provide empirical verifications to the theoretic prediction on 2-layer neural networks, matrix sensing and ResNet-18.

**Summary Of The Review:**

This work studies an important problem (finding better momentum values), and discovers a new optimal momentum scaling under the analysis framework. The practical implication seems to be weak unless the authors could further elaborate.

---

> ### Author Response · Authors · 2022-11-19
> **Response to Reviewer oJpk**
>
> We thank the referee for their positive feedback on our paper, and for the constructive questions that helped expanding and improving our presentation. We believe we addressed all the referee's questions. We detail our reply and updates to the draft below:
>
> 1. **The effect of momentum:** We appreciate this question from the referee. We have updated the introduction to explain more intuitively the effect of momentum. In short, momentum enhances the motion along the zero-loss manifold more than the motion perpendicular to it because the motion along the manifold has a finite expectation while the motion perpendicular to it has zero mean.
>
>     In the interest of completeness, we also now mention the intuition for momentum ($\beta \ne 0$) in the strictly deterministic setting in which it was originally introduced, and in which it is known to have provable convergence benefits. This can be found in the current draft under Related Works, Sec. 2. To summarize briefly, by accumulating gradients as a weighted average, momentum promotes motion not strictly following the local gradient, but moving rather along the directions which tend persistently decrease the loss function across iterations.
>
> 2. **Empirical Work:**  We thank the reviewer for their interest in the practical implications of our result. We performed an experiment with a deep fully connected MLP on the Fashion MNIST dataset to show agreement with our theory on another data-model pair. We present the preliminary data in appendix A.4, showing consistency with our theory. Because of our focus on the phase of training which happens after the training loss approaches zero, it is outside the scope of this work to perform end-to-end comparisons with the state-of-the-art.
>
> 3. **Nature of label noise:** We appreciate this guiding question from the referee and have included more motivation in the paper for using label noise. This type of noise can have empirical benefits. A general situation is when the model fully overfits the data, in which case SGD noise vanishes. For example, when using label noise in matrix sensing, the training loss quickly reaches zero indicating overfitting and, despite this, at later times the test loss will keep decreasing to small values, indicating the benefit of label noise
> [1,2].
>
> 4. **The scaling of momentum:** Our parametrization of $\beta$ is motivated by the following reason. As estimated in Sec. 1.1, in the relevant regime, $\tau_1 = \Theta((1-\beta)^{-1})$ and $\tau_2 = \Theta((1-\beta)^2 \eta^{-2})$. Maximum acceleration is achieved when $\tau_1 = \tau_2$, and solving for $\beta$ as a function of $\eta$ gives the scaling $1-\beta \propto \eta^{2/3}$. Therefore it is natural to parametrize $\beta=1-C\eta^\gamma$ by considering other exponents in this scaling. Intuitively this agrees with our expectation that, as the learning rate decreases, the momentum parameter will have to come closer to 1 in order to improve speed-up, otherwise, if we keep $\beta$ fixed to a value far from 1, the memory of the gradient induced by momentum (discussed in answer 1 above) will only contain increasingly local information about the geometry of the loss landscape surrounding the location of the weights, diminishing the effect of momentum. We have updated our introduction of the scaling law in the paper in accordance with this answer to improve the presentation.
>
> References:
>
> [1] Blanc, Guy, et al. "Implicit regularization for deep neural networks driven by an ornstein-uhlenbeck like process." Conference on learning theory. PMLR, 2020.
>
> [2] Li, Zhiyuan, Tianhao Wang, and Sanjeev Arora. "What Happens after SGD Reaches Zero Loss?--A Mathematical Framework." arXiv preprint arXiv:2110.06914 (2021).

---

> > ### Author Response · Authors · 2022-12-05
> > **Happy to provide additional clarification**
> >
> > We hope our response clarified your initial concerns and questions. We would be happy to provide further clarifications where necessary.

---

### Author Response · Authors · 2022-11-19
**Summary of Changes**

We thank all the reviewers for their constructive feedback, which was especially helpful to reorganize the presentation of our problem setup. We incorporated two main changes:
- **Restructured the heuristic presentation of our result by clarifying Sec. 1.1 and introducing a new Sec. 1.2.**
- **Performed additional experiments and found further confirmation of the scaling we predict.**


Below we provide a list of the changes that we have made to address the points raised.

1. Improved the presentation of the heuristic analysis in Sec. 1.1: we put more emphasis on the assumptions and the key logical implications, breaking equations into smaller steps and using more words to explain the significance of each step.
2. Added Sec. 1.2, which provides an informal overview of the framework used in Secs. 3.1 and 3.2, and re-states our assumptions in the language of that framework.
3. Moved the definition of notations to the paragraph above Sec. 1.1.
4. Added a paragraph at the beginning of Sec. 3.3, stating the rationale of why the noiseless dynamics is sufficient to determine the time $\tau_1$ associated to displacements transverse to the zero-loss manifold.
5. Moved the experimental section on matrix sensing (former Sec. 4.2) to Appendix A.2 due to the length constraint.
6. Added Fig. 4 in Appendix A.3 that shows the performance in test accuracy of Resnet-18 on Cifar10 as we vary the momentum hyperparameter $\beta$. This displays a non-monotonic dependence on $\beta$ and that speed-up is optimal for a particular value of $\beta$, consistent with our prediction.
7. Added experiment using a 6-layer MLP on FashionMNIST in Appendix A.4. We find consistency with our proposed scaling.
8. Added proof of equivalence between eq. (1) and eq. (4) in Appendix B.2.
9. Added a derivation of the variance of the displacements transverse to the zero-loss manifold in Appendix F. This result is now mentioned in Sec. 1.1.
10. Included minor edits to the abstract.

---

> ### Author Response · Authors · 2022-11-29
> **Correction**
>
> We noticed that the new subsection 1.2 contains typos around eq. (3) which we will update in the camera-ready version. We include the corrected version here for the convenience of the referees:
>
> $\ $
>
> Let us collectively denote $x_k=(\pi_k, w_k)$ and write eq. (1) as $x_{k+1}=x_k+F(x_k)+\epsilon \tilde\sigma(x_k)\xi_k$. We can perform a Taylor expansion in $\epsilon$ to obtain $\Phi(x_{k+1})-\Phi(x_k)=\partial\Phi(x_k)[\epsilon \tilde\sigma(x_k)\xi_k]+\frac 12\partial^2\Phi(x_k)[\epsilon \tilde\sigma(x_k)\xi_k,\epsilon \tilde\sigma(x_k)\xi_k]+\cdots$. Therefore, denoting $Y(t=\epsilon^2 k)=\Phi(x_k)$, the limit dynamics as $\epsilon\to 0$ can be well-approximated by the continuous time equation
>
> $dY= \partial\Phi(Y)[ \tilde\sigma(Y)dW] +\frac 12\partial^2\Phi(Y)[ \tilde\sigma(Y) \tilde\sigma(Y)^\top]dt,$
>
> where we interpreted the time increment $dt=\epsilon^2$, and introduced a rescaled noise satisfying $\langle dW^2\rangle=dt$. In the arguments of the functions on the right-hand side, we replaced $x_k$ with $Y(t)$ as intuitively, for $x_k$ very close to $\Gamma$, i.e. $d(x_k,\Gamma)\to 0$ for any $k>0$ as $\epsilon\to 0$, we have $\Phi(x_k)\approx x_k$. Also, we assumed that $dW^2=dt$ up to corrections that are subleading in $dt$.

---

### Decision · Program_Chairs · 2023-01-20

**Decision:**

Reject

**Justification For Why Not Higher Score:**

Look into Weaknesses + Recommendation in the meta-review

**Justification For Why Not Lower Score:**

N/A

**Metareview: Summary, Strengths And Weaknesses:**

- Summary:
This paper analyzes the SDE for SGD with momentum and label noise and tries to show that adding momentum accelerates the convergence but not hurting the generalization. This is an important challenge while training big models with numerous training datasets such as DNN. They show that there is an interplay between the speedup of momentum and the limiting diffusion generated by SGD noise. In particular, this paper studies the convergence of momentum SGD at the vicinity of global minima in the over-parameterized setting. With some approximations which only hold in this vicinity, the scaling analysis predicts an exponent $\gamma = 2/3$  as the scaling of the momentum parameter that maximally accelerates the training.

- Strengths:

1. Interesting SDE-based analysis of SGD for DNNs;
2. The discovery of the optimal momentum scaling is an interesting topic; the holds for the topic of connecting momentum scaling to finding flatter minima;

- Weaknesses:

After the rebuttal discussion, some concerns remain:

1. Overall, any algorithmic modification should be strongly supported by thorough experimental evaluations. Using shallow NNs on small datasets is not sufficient to support the claim that scaling momentum as suggested should be used for large-scale, large-dataset scenarios. While we acknowledge that access to such hardware is not always available, smaller but complicated architectures (like ResNets, CNNs, etc) and different datasets (computer vision, NLP, etc) could be used to validate the momentum scaling. Currently, the experiments considered might validate some of the claims, but the outcome of the paper might be used for other settings with no guaranteed performance.

2. After discussions, some comments by reviewers remain unresolved:

- It is questionable to take the limit of large minibatch size in SGD to control the stochastic noise.
- Theoretically speaking, the theory assumes momentum be exactly zero, so as long as momentum is non-zero (regardless of how small it is), the analysis does not apply.
- Improved presentation of the paper (the paper went through a major revision, which makes reviewing difficult).
- More explanations/ablation studies why the specific scaling is working.



Recommendation:

First of all, the authors should be stayed assured that there were discussions regarding the paper, and the area chair read carefully i) the paper, ii) the reviews, and iii) the discussions after the rebuttal. We acknowledge that the paper addresses concerns raised, but the reviewers have remaining concerns:

We recommend the authors include a more thorough experimental validation for their methodology: including experiments that are common in various fields should be sufficient to answer any questions about the empirical validation of the method. A cleaner support of the theoretical supports (both with theory and practice) is required.

We recommend the authors to consider these (common over all reviewers) concerns to revise their paper; we strongly support for a resubmission to a near-future ML conference.